# Fluctuations of the Largest Eigenvalues of Transformed Spiked Wigner Matrices

**Aro Lee** [1]  **Ji Oon Lee** [1]

## Abstract

We consider a spiked random matrix model obtained by applying a function entrywise to a signal-plus-noise symmetric data matrix. We prove that the largest eigenvalue of this model, which we call a transformed spiked Wigner matrix, exhibits Baik–Ben Arous–Péché (BBP) type phase transition. We show that the law of the fluctuation converges to the Gaussian distribution when the effective signal-to-noise ratio (SNR) is above the critical number, and to the GOE Tracy–Widom distribution when the effective SNR is below the critical number. We provide precise formulas for the limiting distributions and also concentration estimates for the largest eigenvalues, both in the supercritical and the subcritical regimes.

## 1. Introduction

First introduced in the early 1900s, the principal component analysis (PCA) has been widely used as a fundamental method for analyzing multivariate data across a wide range of scientific fields including genetics, economics, and various other disciplines. One notable application of PCA is detecting and recovering a signal from matrix-type data that includes inevitable noise, commonly referred to as signal-plus-noise data.

When the signal is a low-rank matrix, the signal-plus-noise data can be modeled by spiked random matrix models, which play a crucial role in analyzing for many machine learning problems. For instance, in the study of the feature learning in a two-layer neural network, a rank-1 perturbation of the initial weight matrix can approximate the updated weight after one gradient step (Ba et al., 2022). This idea is further developed by (Cui et al., 2024; Dandi et al., 2025), where the spiked Random Features model was considered

with first layer weights. Additionally, these models have practical applications in machine learning algorithms, such as an ensemble method for Q-learning in reinforcement learning (Lee et al., 2023a).

One of the most natural models for signal plus-noise data is a *spiked Wigner matrix*, which is of the form

$$M = W + \sqrt{\lambda} x x^T, \tag{1}$$

where the noise $W$ is an $N \times N$ symmetric (real) Wigner matrix, $x \in \mathbb{R}^N$ is the spike, and $\lambda \in \mathbb{R}$ is the signal-to-noise ratio (SNR). (See Definitions 2.1 and 2.2.) Spiked Wigner matrices appear naturally in inference problems where the data is obtained from pairwise measurements. Notable examples of such problems include the community detection from the stochastic block model and synchronization over $\mathbb{Z}/2$.

It is well-known that the largest eigenvalue of $M$ exhibits a phase transition, known as the BBP transition named after the seminal work by Baik, Ben Arous, and Péché (Baik et al., 2005), depending on the parameter $\lambda$. If $\lambda > 1$, then the largest eigenvalue of $M$ converges to $\sqrt{\lambda} + \frac{1}{\sqrt{\lambda}}$, separated from the other eigenvalues of $M$. On the other hand, if $\lambda < 1$, the largest eigenvalue converges to 2, which is the spectral edge of the pure noise model.

In this paper, we consider an entrywise transformation of the spiked Wigner matrix in (1), defined by

$$\widetilde{M}_{ij} = N^{-\frac{1}{2}} f(\sqrt{N} M_{ij}) \tag{2}$$

for a function $f$. (Note that a typical size of the entries of a Wigner matrix is of order $N^{-1/2}$.) Such an entrywise transformation was considered in (Lesieur et al., 2015; Perry et al., 2018) to improve PCA by effectively increasing the SNR. Heuristically, the entries of the transformed spiked Wigner matrix in (2) can be approximated using the Taylor expansion by

$$
\begin{aligned}
N^{-\frac{1}{2}} f(\sqrt{N} M_{ij}) &= N^{-\frac{1}{2}} f(\sqrt{N} W_{ij} + \sqrt{\lambda N} x_i x_j) \\
&\approx \frac{f(\sqrt{N} W_{ij})}{\sqrt{N}} + \sqrt{\lambda} f'(\sqrt{N} W_{ij}) x_i x_j \\
&\approx \frac{f(\sqrt{N} W_{ij})}{\sqrt{N}} + \sqrt{\lambda} \mathbb{E}[f'(\sqrt{N} W_{ij})] x_i x_j.
\end{aligned}
\tag{3}
$$

[1]Department of Mathematical Sciences, KAIST, Daejeon, Korea. Correspondence to: Aro Lee <sditar444@kaist.ac.kr>, Ji Oon Lee <jioon.lee@kaist.edu>.

*Proceedings of the 42$^{nd}$ International Conference on Machine Learning*, Vancouver, Canada. PMLR 267, 2025. Copyright 2025 by the author(s).

The transformed matrix $\widetilde{M}$ is approximately a spiked Wigner matrix, where the SNR is changed to $\lambda(\mathbb{E}[f'(\sqrt{N}W_{ij})])^2$. By optimizing $f$, it is possible to make the new SNR larger than the original SNR, which makes the PCA with the transformed matrix (called the transformed PCA) stronger than the vanilla PCA. The approximation in (3) is proven to be valid in (Perry et al., 2018), in the sense that the difference between the largest eigenvalues of $\widetilde{M}$ and its approximation defined by the right side of (3) is $o(1)$. We note that, while Theorem 4.8 in (Perry et al., 2018) considers the approximation in (3) only for the optimal transformation, the same proof works for any transformation.

Suppose now that we want to detect the presence of the signal in the given data matrix by PCA. It would require to compute the largest eigenvalues of the data matrix, estimate the $p$-value from the known limiting distributions of the largest eigenvalues, and compare it with a predefined significance level; for an algorithm involving the largest eigenvalue only, see (Kritchman & Nadler, 2008), and for an algorithm based on the ratio between the top eigenvalues, see (Onatski, 2009). If the given data matrix is a spiked Wigner matrix, the limiting distributions of the largest eigenvalues are also well-known, which is also a part of the BBP transition; if $\lambda > 1$, then the fluctuation of the largest eigenvalue of $M$ is given by a Gaussian of order $N^{-1/2}$, whereas if $\lambda < 1$, it is given by GOE Tracy–Widom distribution of order $N^{-2/3}$. However, a corresponding result for transformed spiked Wigner matrices are not known.

## 1.1. Contributions

In this paper, we prove the fluctuations of the largest eigenvalue of the transformed spiked Wigner matrix $\widetilde{M}$ coincide with those of the spiked Wigner matrix. More precisely, under mild assumptions on $\widetilde{M}$ (see Assumptions 2.3 and 2.4) if we let $\mu_1$ be the largest eigenvalue of $\widetilde{M}$ and the effective SNR $\lambda_e = \lambda(\mathbb{E}[f'(\sqrt{N}W_{12})])^2$, then

- (Supercritical case) If $\lambda_e > 1$, then $N^{1/2}(\mu_1 - (\sqrt{\lambda_e} + \frac{1}{\sqrt{\lambda_e}}))$ converges to a Gaussian distribution with mean 0 and variance $2(\lambda_e - 1)/\lambda_e$.

- (Subcritical case) If $\lambda_e < 1$, then $N^{2/3}(\mu_1 - 2)$ converges to the GOE Tracy–Widom distribution.

- (Rigidity) For both the supercritical and the subcritical cases, the deviation of the largest eigenvalue from its limit cannot be significantly larger than the typical size of the fluctuation with overwhelming probability.

See Theorems 2.5 and 2.6 for the precise statements. This establishes the BBP transition for the largest eigenvalue of

$\widetilde{M}$. We also consider some specific examples to compare the numerical results with the theoretical results.

The main technical difficulty in the proof is that the error terms in the approximation in (3) is not negligible a priori. Most precisely, in (3), the term

$$\sqrt{\lambda}\left(f'(\sqrt{N}W_{ij}) - \mathbb{E}[f'(\sqrt{N}W_{ij})]\right)x_i x_j \qquad (4)$$

is ignored. However, if we follow the analysis in (Perry et al., 2018), the norm of the matrix whose entries are given by (4) is of order $N^{-1/2}$, which is of the same size as the fluctuation of the largest eigenvalue of a spiked Wigner matrix in the supercritical case, and much larger than that of the largest eigenvalue in the subcritical case. It is thus impossible to ignore the term in (4), but then the noise matrix is no more a Wigner matrix since the size of $x_i$ and $x_j$ may not be uniform. Another issue is that the second derivative term in the Taylor expansion in (3) is not negligible, since it is also of order $N^{-1/2}$ a priori. Thus, the correct approximation should be not by a (rank-1) spiked Wigner matrix but a rank-2 spiked Wigner-type matrix. (See Proposition 3.1 for more detail.)

To prove the BBP transition for the rank-2 spiked Wigner-type matrix, we apply the Green function comparison argument. In this strategy, the distribution function of the largest eigenvalue is first approximated by a functional of the resolvent of the matrix. Then, by comparing the resolvent of the given matrix and that of the target matrix for which the limiting distribution of the largest eigenvalue is known, it is possible to prove that the fluctuations of the largest eigenvalues of two matrices are equal. The target matrices are a rank-2 spiked Wigner matrix for the supercritical case, and a Wigner matrix for the subcritical case. The Green function comparison argument is conducted by following a continuous matrix flow, typically called an interpolation, and it requires a technical input, known as the local law, which is the estimate on the resolvent of the matrices.

The rigidity of the largest eigenvalue is one of the main technical inputs for the Green function comparison argument, and it is also of separate interest since it asserts strong concentration of the largest eigenvalue in near-optimal scale. In the subcritical case, the rigidity is a consequence of a stronger estimate on the difference between the largest eigenvalues of the noise matrix with and without a spike, known as the eigenvalue sticking.

## 1.2. Applications

Our results can provide theoretical background for the understanding of transformed spiked Wigner matrices, especially (asymptotically) exact error probability of the improved PCA for the finite $N$ case. We list possible examples where transformed spiked Wigner matrices are useful.

**Improved PCA.** It was proved in (Perry et al., 2018) that PCA achieves the optimal strong detection threshold for a spiked Wigner matrix with Gaussian noise, in the sense that it is impossible to reliably distinguish (with probability $1 - o(1)$) a Wigner matrix with a spike and one without a spike if $\lambda < 1$. The threshold for the strong detection is lowered if the noise is non-Gaussian, but it can be matched by improving PCA via entrywise transformation, with the function $f = -p'/p$ where $p$ is the density of the noise.

**Machine learning theory.** In deep neural networks, the transformed spiked Wigner matrix can be used in theoretical analysis, where the noise corresponds to the pre-activation, the spike is due to initial training, and the entrywise transformation is the activation function. Another important example is the feature matrix of the two-layer neural network, where the SNR corresponds to the step size.

### 1.3. Related Works

The spiked random matrix model was introduced by Johnstone (Johnstone, 2001) for Wishart matrices (Gaussian i.i.d. rectangular matrices) with spiked covariance. The transition of the largest eigenvalue was proved by Baik, Ben Arous, and Péché (Baik et al., 2005) for spiked complex Wishart matrices and generalized to other models, including the spiked Wigner matrices (Péché, 2006; Féral & Péché, 2007; Capitaine et al., 2009; Benaych-Georges & Nadakuditi, 2011; Knowles & Yin, 2013; Pizzo et al., 2013; Renfrew & Soshnikov, 2013; Knowles & Yin, 2014; Bloemendal et al., 2016). The detection problems for the spiked Wigner matrices have been extensively studied recently, and many important results, including the theoretical limits (Perry et al., 2018; El Alaoui et al., 2020; Chung et al., 2025; Moitra & Wein, 2025) and algorithms (Perry et al., 2018; Chung & Lee, 2019; 2022), have been proved. For more results on the spiked Wigner matrices, we refer to (Moitra & Wein, 2025) and references therein.

The spectral properties of spiked random matrix models are important in analyzing various problems in machine learning and statistics, as spectral methods can improve convergence analysis and provides theoretical guarantees. Notable examples include the community detection (Abbe, 2017) and submatrix localization (Butucea et al., 2013). In the context of machine learning, we refer to (Mondelli & Venkataramanan, 2021) for an application to the approximate message passing algorithm and (Chi et al., 2019) to non-convex optimization.

Entrywise transformed random matrix models provide valuable insights into the study of fundamental limits of detection for the signal from a spiked random matrix, which has been extensively studied by analyzing the eigenvalues (Montanari et al., 2015; Johnstone & Onatski, 2020) or the mutual information and minimum mean squared error

(Lesieur et al., 2015; Krzakala et al., 2016; Dia et al., 2016; Lelarge & Miolane, 2019). Entrywise transformation of spiked Wigner matrices was first described by Lesieur, Krzakala, and Zdeborová (Lesieur et al., 2015) and rigorously analyzed by Perry, Wein, Bandeira, and Moitra (Perry et al., 2018). The transition of the limit of the largest eigenvalue of transformed spiked Wigner matrices was first proved in (Perry et al., 2018), where the correlation between the top eigenvector and the spike was also proved. Similar results were proved with weaker assumptions (Guionnet et al., 2023; Feldman, 2025; Moniri & Hassani, 2024) and also generalized to other models (Jung et al., 2021; Guionnet et al., 2023; Feldman, 2025; Mergny et al., 2024), including spiked rectangular matrices.

The spiked random matrix models and their entrywise transformation have been extensively used in the theoretical study of neural networks. These models contribute to the theoretical analysis of deep neural network, where entrywise transformations correspond to pointwise nonlinear activation functions (Pennington & Worah, 2017). They are also crucial in the analysis on the feature learning. See, e.g., (Damian et al., 2022; Ba et al., 2022; Lee et al., 2023b; Mousavi-Hosseini et al., 2023; Moniri et al., 2024; Cui et al., 2024; Ba et al., 2024). For more results, we refer to (Cui et al., 2024) and references therein.

### 1.4. Organization of the Paper

The rest of the paper is organized as follows: In Section 2, we precisely define the model and state our main results. In Section 3 and Section 4, we outline the proofs of our main results for the supercritical case and the subcritical case, respectively. In section 5, we conduct numerical experiments to compare theoretical results with numerical results for several examples. We conclude the paper in Section 6 with a summary of our results and directions for future research. Details of the numerical simulations and the proofs of the results can be found in Appendices.

*Notational Remark.* We use the standard big-O and little-o notation. For an event $\Omega$, we say that $\Omega$ holds with overwhelming probability if for any (large) $D > 0$ there exists $N_0 \equiv N_0(D)$ such that $\mathbb{P}(\Omega^c) < N^{-D}$ whenever $N > N_0$. For a sequence of random variables, the notation $\Rightarrow$ denotes the convergence in distribution as $N \to \infty$.

## 2. Main Result

### 2.1. Definition of the Model

We first define the model we consider in this paper. We assume that the noise matrix is a Wigner matrix for which we use the following definition.

**Definition 2.1** (Wigner matrix). We say an $N \times N$ random matrix $W = (W_{ij})$ is a Wigner matrix if $W$ is symmetric

and $W_{ij}$ ($1 \leq i \leq j \leq N$) are independent real random variables satisfying the following conditions:

- For all $i, j$, $\mathbb{E}[W_{ij}] = 0$ and for any ($N$-independent) positive integer $D$, $N^{D/2}\mathbb{E}[|W_{ij}|^D] \leq C_D$ for some ($N$-independent) constant $C_D$.

- For all $i < j$, $N\mathbb{E}[|W_{ij}|^2] = 1$.

Note that the upper triangle entries of a Wigner matrix are not necessarily identically distributed. The signal-plus-noise model we consider is a (rank-1) spiked Wigner matrix, which is defined as follows:

**Definition 2.2** (Spiked Wigner matrix). We say an $N \times N$ random matrix $M = W + \sqrt{\lambda}xx^T$ is a spiked Wigner matrix with a spike $x$ and signal-to-noise ratio (SNR) $\lambda$ if $W$ is a Wigner matrix and $x = (x_1, x_2, \ldots, x_N) \in \mathbb{R}^N$ with $\|x\|_2 = 1$.

Throughout the paper, we assume that $\lambda$ is independent of $N$. (See Remark 2.8 for the discussion on the case $\lambda \gg 1$.) In addition, we assume the following for the data matrix.

**Assumption 2.3.** Let $M = W + \sqrt{\lambda}xx^T$ be a spiked Wigner matrix as in Definition 2.2. For the spike $x = (x_1, x_2, \ldots, x_N)$, we assume the following:

- For any $\epsilon > 0$, $\max_i |x_i| = O(N^{-\frac{1}{2}+\epsilon})$, $\sum_i x_i = O(N^\epsilon)$, and $\sum_i x_i^3 = O(N^{-1+\epsilon})$.

For the noise matrix $W$, we assume that the following:

- For all $i \leq j$, the normalized entries $\sqrt{N}W_{ij}$ are identically distributed.

- For all $i, j$ and any fixed $D$, the $D$-th moment of $\sqrt{N}W_{ij}$ is finite.

We remark that the assumption on the spike is satisfied with the i.i.d. prior, where we let $y_i$ be i.i.d. with an $N$-independent distribution whose mean is $0$, variance $1$, and all moments are finite, and set $x_i = y_i/\sqrt{N}$. The assumption is also satisfied for many other priors such as the spherical prior. We believe our result can also be proved with weaker assumptions. Especially, the same result would hold even when the distribution of the diagonal entries are different from that of the off-diagonal entries. However, we do not pursue it in the current paper.

Lastly, we assume that the entrywise transformation satisfies the following properties.

**Assumption 2.4.** For a given Wigner matrix $W$, we assume that the following holds with the function $f : \mathbb{R} \to \mathbb{R}$.

- The function $f$ is $C^3$ and its derivatives are polynomially bounded in the sense that $|f^{(\ell)}(w)| \leq C_\ell(1 + |w|)^{C_\ell}$ ($\ell = 0, 1, 2, 3$) for some constant $C_\ell$.

- For all $i$ and $j$, $\mathbb{E}[f(\sqrt{N}W_{ij})] = 0$. Furthermore, $\mathbb{E}[f(\sqrt{N}W_{12})^2] = 1$ and $\mathbb{E}[f'(\sqrt{N}W_{12})] \geq 0$.

We remark that the last part of Assumption 2.4 is not restrictive, since it can be satisfied by simply multiplying the function $f$ by a suitable constant.

## 2.2. Main Result

Our main result is the following theorem on asymptotic normality of the largest eigenvalue of the transformed matrix.

**Theorem 2.5.** *Suppose that $M$ is a spiked Wigner matrix, satisfying Assumption 2.3. Let $\widetilde{M}$ be a matrix defined by $\widetilde{M}_{ij} = N^{-\frac{1}{2}}f(\sqrt{N}M_{ij})$ for a function $f$ satisfying Assumption 2.4. Let $\mu_1(\widetilde{M})$ be the largest eigenvalue of $\widetilde{M}$. Set*

$$\lambda_e := \lambda(\mathbb{E}[f'(\sqrt{N}W_{12})])^2. \tag{5}$$

- *(Supercritical case) If $\lambda_e > 1$, then*

$$N^{1/2}\left(\mu_1(\widetilde{M}) - \left(\sqrt{\lambda_e} + \frac{1}{\sqrt{\lambda_e}}\right)\right) \Rightarrow \mathcal{N}(0, \sigma^2), \tag{6}$$

*where the right-hand side of (6) is a Gaussian distribution with mean $0$ and variance*

$$\sigma^2 := \frac{2(\lambda_e - 1)}{\lambda_e} = \frac{2(\lambda(\mathbb{E}[f'(\sqrt{N}W_{12})])^2 - 1)}{\lambda(\mathbb{E}[f'(\sqrt{N}W_{12})])^2}. \tag{7}$$

- *(Subcritical case) If $\lambda_e < 1$, then*

$$N^{2/3}\left(\mu_1(\widetilde{M}) - 2\right) \Rightarrow TW_1, \tag{8}$$

*where the right-hand side of (8) is the GOE Tracy–Widom distribution.*

Theorem 2.5 shows that our model, the transformed spiked Wigner matrix, exhibits the BBP transition, and it coincides with that of the (non-transformed) spiked Wigner matrix even in terms of the fluctuation with $\lambda_e$ being the effective SNR.

In addition to Theorem 2.5, we also have the following result on the rigidity of the largest eigenvalue.

**Theorem 2.6.** *Suppose that the assumptions in Theorem 2.5 hold. Then, for any $\epsilon > 0$, the following holds with overwhelming probability.*

- *(Supercritical case) If $\lambda_e > 1$, then*

$$\mu_1(\widetilde{M}) - \left(\sqrt{\lambda_e} + \frac{1}{\sqrt{\lambda_e}}\right) = O(N^{-\frac{1}{2}+\epsilon}). \tag{9}$$

- *(Subcritical case) If $\lambda_e < 1$, then*

$$\mu_1(\widetilde{M}) - 2 = O(N^{-\frac{2}{3}+\epsilon}). \tag{10}$$

For the supercritial case, Theorem 2.6 was essentially proved in (Perry et al., 2018). For the sake of completeness, we will prove it in Appendices. The rigidity of the largest eigenvalue for the subcritical case is a consequence of a stronger result, known as the eigenvalue sticking. For a spiked Wigner matrix in the subcritical case, it means that the difference between the largest eigenvalue of a Wigner matrix $W$ and that of the spiked Wigner matrix $W + \sqrt{\lambda} \boldsymbol{x} \boldsymbol{x}^T$ is much less than $N^{-2/3}$, which is the size of the fluctuation of the largest eigenvalue. (See, e.g., Theorem 2.7 in (Knowles & Yin, 2013).)

*Remark* 2.7. Adapting the strategy for the proof of Theorems 2.5 and 2.6, we believe that it is also possible to prove corresponding statements for transformed rank-$k$ spiked Wigner matrices for any fixed $k \geq 2$. A rank-$k$ spiked Wigner matrix is of the form

$$M = W + \sum_{i=1}^{k} \sqrt{\lambda^{(i)}} \boldsymbol{x}^{(i)} (\boldsymbol{x}^{(i)})^T, \tag{11}$$

where $W$ is an $N \times N$ Wigner matrix, $\boldsymbol{x}^{(i)} \in \mathbb{R}^N$ with $\langle \boldsymbol{x}^{(i)}, \boldsymbol{x}^{(j)} \rangle = \delta_{ij}$, and $\lambda^{(1)} > \cdots > \lambda^{(k)} > 0$. Let $\widetilde{M}$ be the transformed matrix defined by $\widetilde{M}_{ij} = N^{-\frac{1}{2}} f(\sqrt{N} M_{ij})$ and let $\lambda_e^{(i)} = \lambda^{(i)} (\mathbb{E}[f'(\sqrt{N} W_{12})])^2$. Suppose that

$$\lambda_e^{(1)} > \cdots > \lambda_e^{(\ell)} > 1 > \lambda_e^{(\ell+1)} > \cdots > \lambda_e^{(k)} > 0.$$

It was proved in (Jung et al., 2024) that, almost surely, (1) if $i \leq \ell$, then $\mu_i(\widetilde{M}) \to \sqrt{\lambda_e^{(i)} + \frac{1}{\sqrt{\lambda_e^{(i)}}}}$ and (2) if $i > \ell$, then $\mu_i(\widetilde{M}) \to 2$. (While Theorem 1 in (Jung et al., 2024) is stated only for the optimal transformation of the form $-p'/p$, the same proof works for any transformation.) The BBP transition for $\widetilde{M}$ would assert that (1) if $i < \ell$, then $N^{1/2} \left( \mu_i(\widetilde{M}) - \left( \sqrt{\lambda_e^{(i)} + \frac{1}{\sqrt{\lambda_e^{(i)}}}} \right) \right)$ converges in distribution to a centered Gaussian with variance $2(\lambda_e^{(i)} - 1)/\lambda_e^{(i)}$ and (2) if $i > \ell$, then the fluctuation of $\mu_i(\widetilde{M})$ coincides with that of the $(i - \ell)$-th largest eigenvalue of a GOE matrix. However, to avoid complication, we refrain from considering the rank-$k$ models in this paper.

*Remark* 2.8. If $\lambda_e = 0$ in Theorem 2.5, it is natural to consider a different scaling of the SNR $\lambda$ for the BBP transition. Suppose that $\mathbb{E}[f'(\sqrt{N} W_{12})] = 0$ and $\mathbb{E}[f''(\sqrt{N} W_{12})] \neq 0$. With the scaling $\lambda \equiv \lambda(N) = \lambda_0 \sqrt{N}$, the Taylor expansion in (3) is then

$$N^{-\frac{1}{2}} f(\sqrt{N} M_{ij})$$
$$\approx \frac{f(\sqrt{N} W_{ij})}{\sqrt{N}} + \frac{1}{2} \lambda_0 N \mathbb{E}[f''(\sqrt{N} W_{ij})] x_i^2 x_j^2.$$

This suggests that the effective SNR

$$\widetilde{\lambda}_e = (\lambda_0^2/4) N^2 \mathbb{E}[f''(\sqrt{N} W_{12})]^2 \|\boldsymbol{x}^2\|^4,$$

where $\boldsymbol{x}^2 = (x_1^2, x_2^2, \ldots, x_N^2)$. Notice that for the i.i.d. prior, $\|\boldsymbol{x}^2\|^2 = N^{-1} \mathbb{E}[x_1^4] + o(N^{-1})$.

In (Guionnet et al., 2023), it was proved that the limit of the largest eigenvalue exhibits the BBP transition with the effective SNR $\widetilde{\lambda}_e$. Adapting the strategy for the proof of Theorems 2.5 and 2.6, it is also possible to prove that the corresponding statements for the fluctuation of the largest eigenvalue. We remark that the actual statement also contains a deterministic shift term of order $N^{-1/2}$; see (79) for more precise statement. In Appendix D, we provide the idea of the proof for this case. It can be further generalized by considering

$$k_f := \inf\{k \in \mathbb{Z}^+ : \mathbb{E}[f^{(k)}(\sqrt{N} W_{ij})] \neq 0\}$$

and the scaling $\lambda = \lambda_0 N^{\frac{1}{2}(1 - \frac{1}{k_f})}$. See Appendix D.4 for more discussion.

## 3. Outline of the Proof - Supercritical Case

In this section, we outline the proof of the first part of our main result, Theorem 2.5. (The detailed proofs for the results in Section 3 can be found in Appendix B.) Throughout Section 3, we assume $\lambda_e > 1$.

### 3.1. Approximation by a Spiked Random Matrix

We begin the proof of Theorem 2.5 by approximating the transformed matrix $\widetilde{M}$ by a spiked random matrix. As discussed in Introduction, it is required to approximate $\widetilde{M}$ by a spiked Wigner-type matrix instead of a spiked Wigner matrix, which is obtained from the Taylor expansion for the entrywise transformation $f$. The first step of this approximation is the following proposition:

**Proposition 3.1.** *Suppose that $M$ is a spiked Wigner matrix satisfying Assumption 2.3. Let $\widetilde{M}$ be a matrix defined by $\widetilde{M}_{ij} = N^{-\frac{1}{2}} f(\sqrt{N} M_{ij})$. Define an $N \times N$ random matrix $H$ by*

$$H_{ij} = \frac{f(\sqrt{N} W_{ij})}{\sqrt{N}} + \sqrt{\lambda} f'(\sqrt{N} W_{ij}) x_i x_j \\ + \frac{\lambda}{2} \mathbb{E}[f''(\sqrt{N} W_{ij})] \sqrt{N} x_i^2 x_j^2. \tag{12}$$

*Let $\mu_1(\widetilde{M})$ and $\mu_1(H)$ be the largest eigenvalues of $\widetilde{M}$ and $H$, respectively. Then, for any $\epsilon > 0$, $\mu_1(\widetilde{M}) - \mu_1(H) = O(N^{-1+\epsilon})$ with overwhelming probability.*

Proposition 3.1 basically asserts that in the Taylor expansion for $f$, the terms involving the third or higher derivatives can be ignored. Moreover, the terms involving the second derivative can only affect the largest eigenvalue via its expectation. Note that it is an a priori estimate and we will eventually see that in the Taylor expansion, only the terms

involving the first derivative affect the largest eigenvalue, via its expectation, which justifies the approximation in (3).

Our strategy for the proof of Theorem 2.5 is to compare the largest eigenvalue of $H$ with that of a rank-2 spiked Wigner matrix (in the sense of (11)) for which the asymptotic normality of the fluctuation of the outlier eigenvalue is known. The main obstacle is that the 'noise' matrix in $H$ is not a Wigner matrix. To analyze this more in detail, let us introduce the short-handed notation

$$f_{ij} := f(\sqrt{N}W_{ij}), \quad f'_{ij} := f'(\sqrt{N}W_{ij}),$$
$$f''_{ij} := f''(\sqrt{N}W_{ij}), \quad f^{(k)}_{ij} := f^{(k)}(\sqrt{N}W_{ij}). \tag{13}$$

(From the assumption on the function $f$, we have that $\mathbb{E}[f_{ij}] = 0$ for all $i, j$, and $\mathbb{E}[f^2_{12}] = 1$.) If we denote the noise matrix in $H$ by $V$, whose entries are

$$V_{ij} = \frac{f_{ij}}{\sqrt{N}} + \sqrt{\lambda}(f'_{ij} - \mathbb{E}[f'_{ij}])x_i x_j, \tag{14}$$

then the matrix $V$ is a symmetric random matrix whose upper triangle entries are independent random variables (up to symmetric constraint), satisfying $\mathbb{E}[V_{ij}] = 0$, but

$$\mathbb{E}[(V_{ij})^2] = \frac{1}{N} + \frac{2\sqrt{\lambda}}{\sqrt{N}} \left( \mathbb{E}[f_{ij}f'_{ij}] - \mathbb{E}[f_{ij}]\mathbb{E}[f'_{ij}] \right) x_i x_j$$
$$+ \lambda \left( \mathbb{E}[(f'_{ij})^2] - \mathbb{E}[f'_{ij}]^2 \right) x^2_i x^2_j$$
$$=: \frac{1}{N}(1 + C^V_1 \sqrt{N} x_i x_j + C^V_2 N x^2_i x^2_j). \tag{15}$$

Thus, $V$ is not a Wigner matrix but a Wigner-type matrix.

Although it might seem natural to interpolate between $V_{ij}$ and $f_{ij}/\sqrt{N}$, this approach is complicated by the correlation between $f_{ij}/\sqrt{N}$ and the sub-leading order term $\sqrt{\lambda}(f'_{ij} - \mathbb{E}[f'_{ij}])x_i x_j$. In this paper, we instead consider another interpolation, defined by a matrix $V(t)$ for $t \in [0, 1]$ whose entries are given by

$$V(t)_{ij} := \sqrt{\frac{1 + C^V_1 t\sqrt{N} x_i x_j + C^V_2 tN x^2_i x^2_j}{N\mathbb{E}[(V_{ij})^2]}} V_{ij}. \tag{16}$$

By definition, $V(1) = V$ and $V(t)$ is a Wigner-type matrix for any $t \in [0, 1]$. We can also see that $V(0)$ is a Wigner matrix whose off-diagonal entries are not identically distributed. Note that it does not affect our proof since the fluctuation of the outlier eigenvalue is known for spiked Wigner matrices, even when the entries of the Wigner matrix are not identically distributed.

With $V(t)$, we finally introduce the desired interpolation, defined by a matrix $H(t)$ for $t \in [0, 1]$ with entries

$$H(t)_{ij} = V(t)_{ij} + \sqrt{\lambda}\mathbb{E}[f'_{ij}]x_i x_j + \frac{\lambda}{2}\mathbb{E}[f''_{ij}]\sqrt{N}x^2_i x^2_j. \tag{17}$$

Note that $H(0)$ is a rank-2 spiked Wigner matrix and $H = H(1)$.

## 3.2. Local Law

To estimate the change of the largest eigenvalue along the matrix flow given by $H(t)$, we use the Green function comparison argument. It relies on the fact that the distribution function of the largest eigenvalue of a matrix can be well approximated by a certain functional of the resolvent of the matrix. (See Proposition 3.3 for an example of such a functional.) The main technical input necessary for the Green function comparison is an estimate for the resolvent of the matrix, known as the local law. We will use the following local law for the resolvent of $H(t)$.

**Proposition 3.2.** *Let $H(t)$ be the matrix defined in (17). Define the resolvent of $H(t)$ by*

$$G(t, z) := (H(t) - zI)^{-1} \quad (\text{Im } z \geq 0). \tag{18}$$

*Set $z \equiv \tau + i\eta$ and*

$$\kappa := \tau - \left( \sqrt{\lambda}\mathbb{E}[f'_{12}] + \frac{1}{\sqrt{\lambda}\mathbb{E}[f'_{12}]} \right).$$

*Suppose that $|\kappa| \leq N^{-\frac{1}{2}+\epsilon}$ and $\eta = N^{-\frac{1}{2}-\epsilon}$ for a fixed $\epsilon > 0$. Then,*

$$\max_{i,j} |G_{ij}(t, z) - m_{sc}(z)\delta_{ij}| = \mathcal{O}(N^{-\frac{1}{2}+6\epsilon}), \tag{19}$$

*uniformly on $t \in [0, 1]$, where $m_{sc}$ is the Stieltjes transform of the Wigner semicircle law, given by $m_{sc}(z) = (-z + \sqrt{z^2 - 4})/2$.*

For the proof of Proposition 3.2, we first prove a corresponding result for the resolvent of $V(t)$ and then apply some resolvent identities. Define the resolvent of $V(t)$ by

$$R(t, z) := (V(t) - zI)^{-1} \quad (\text{Im } z \geq 0). \tag{20}$$

A local law for $R(t, z)$ was proved in (Ajanki et al., 2017) where $R_{ij}(t, z)$ was approximated by a deterministic number $m_i(t, z)$, given as the solution of the quadratic vector equation

$$-\frac{1}{m_i(t, z)} = z + \sum_j \mathbb{E}[(V(t)_{ij})^2]m_j(t, z). \tag{21}$$

Note that if $\mathbb{E}[(V(t)_{ij})^2] = N^{-1}$ for all $i, j$, then (21) is satisfied by $m_i(0, z) = m_{sc}(z)$ for all $i$, since $-1/m_{sc}(z) = z + m_{sc}(z)$.

For our purpose, the local law involving $m_i(t, z)$ is not sufficient since we need an estimate on $R_{ij}(t, z)$ that is uniform in $i, j$. To obtain such an estimate, we notice that while $V(t)$ is not a Wigner matrix for $t > 0$, it is asymptotically a Wigner matrix in the sense that $\mathbb{E}[(V(t)_{ij})^2] = N^{-1} + O(N^{-\frac{3}{2}+2\epsilon})$. Thus, it is possible to bound the difference $|m_{sc}(z) - m_i(t, z)|$ and prove the desired local law for $R(t, z)$. (See Lemma B.3 for the precise statement.)

### 3.3. Green Function Comparison

With the local law in Proposition 3.2, we can compare functionals of the resolvents of $H(0)$ and $H(1)$. Before the comparison, we first introduce a well-known example of such a functional, in Proposition 3.3. We apply the following three-step procedure: (i) the cutoff of the eigenvalues by an indicator function on a small interval around the limit of the largest eigenvalue (by $\chi_E$ in Proposition 3.3), (ii) mollification of the cutoff function by the Poisson kernel (by $\theta_\eta$ in Proposition 3.3), and (iii) approximation of the distribution function of the largest eigenvalue by the (mollified) cutoff function (by $K$ in Proposition 3.3).

**Proposition 3.3.** *Let $H(t)$ be the matrix defined in (17) and denote by $\mu_1(H(t))$ the largest eigenvalue of $H(t)$. Define*

$$L := \sqrt{\lambda_e} + \frac{1}{\sqrt{\lambda_e}} = \sqrt{\lambda}\mathbb{E}[f'_{12}] + \frac{1}{\sqrt{\lambda}\mathbb{E}[f'_{12}]}$$

*Fix $\epsilon > 0$. Let $E \in \mathbb{R}$ such that $|E - L| \leq N^{-1/2+\epsilon}$. Let $E_+ := L + 2N^{-1/2+\epsilon}$ and define $\chi_E := \chi_{[E,E_+]}$. Set $\eta_1 := N^{-1/2-\epsilon/2}$ and $\eta_2 := N^{-1/2-3\epsilon}$. Let $K : \mathbb{R} \to [0, \infty)$ be a smooth function satisfying*

$$K(x) = \begin{cases} 1 & \text{if } |x| < 1/3 \\ 0 & \text{if } |x| > 2/3 \end{cases}$$

*which is a monotone decreasing function on $[0, \infty)$. Define the Poisson kernel $\theta_\eta$ for $\eta > 0$ by*

$$\theta_\eta(y) := \frac{\eta}{\pi(y^2 + \eta^2)}.$$

*Then, for any $D > 0$,*

$$\mathbb{E}[K(\mathrm{Tr}(\chi_E * \theta_{\eta_2})(H))] > \mathbb{P}(\mu_1(H(t)) \leq E - \eta_1) - N^{-D}$$

*and*

$$\mathbb{E}[K(\mathrm{Tr}(\chi_E * \theta_{\eta_2})(H))] < \mathbb{P}(\mu_1(H(t)) \leq E - \eta_1) + N^{-D}$$

*for any sufficiently large $N$.*

Heuristically, if $\mu_1(H(t)) > E$, then $\mathrm{Tr}(\chi_E * \theta_{\eta_2})(H)$ is nearly 1. By applying $K$, we change it to 0, so that it can approximate the indicator function $\mathbf{1}(\mu_1(H(t)) \leq E)$. Note that we assume $K$ is smooth to ensure that the Green function comparison theorem is applicable. The precise statement of the Green function comparison theorem is as follows.

**Proposition 3.4.** *Let $\epsilon > 0$ and set $\eta = N^{-\frac{1}{2}-\epsilon}$. Let $E_1, E_2 \in \mathbb{R}$ satisfy*

$$|E_\ell - L| \leq N^{-\frac{1}{2}+\epsilon} \quad (\ell = 1, 2).$$

*Let $F : \mathbb{R} \to \mathbb{R}$ be a smooth function satisfying*

$$\max|F^{(m)}(x)| \leq C_m(1 + |x|)^C \qquad (m = 0, 1, 2, 3, 4)$$

*for some constants $C_m > 0$. Then, for any sufficiently small $\epsilon > 0$, there exists $\delta > 0$ such that*

$$\left| \mathbb{E}F\left(\mathrm{Im}\int_{E_1}^{E_2} \mathrm{Tr}\, G(1, x + \mathrm{i}\eta)\mathrm{d}x\right) \right.$$
$$\left. - \mathbb{E}F\left(\mathrm{Im}\int_{E_1}^{E_2} \mathrm{Tr}\, G(0, x + \mathrm{i}\eta)\mathrm{d}x\right) \right| \leq N^{-\delta}. \tag{22}$$

*for any sufficiently large $N$.*

### 3.4. Proof of Theorem 2.5 - Supercritical Case

To finish the proof, we need to compute the eigenvalues of the spike in $H(1)$. Let $\boldsymbol{x}^2 := (x_1^2, x_2^2, \ldots, x_N^2)$. Note that $\boldsymbol{x}$ and $\boldsymbol{x}^2$ are not orthogonal, thus $\sqrt{\lambda_e} = \sqrt{\lambda}\mathbb{E}[f'_{12}]$ may not be an eigenvalue of $H(1) - V(1)$.

**Proposition 3.5.** *Let $A_N := H(t) - V(t) = \sqrt{\lambda_e}\boldsymbol{x}\boldsymbol{x}^T + \frac{\lambda}{2}\mathbb{E}[f''_{12}]\sqrt{N}\boldsymbol{x}^2(\boldsymbol{x}^2)^T$. Denote the ordered eigenvalues of $A_N$ by $\theta_1 \geq \theta_2 \geq \cdots \geq \theta_N$. Then, for any $\epsilon > 0$,*

1. *$\theta_1 = \sqrt{\lambda_e} + O(N^{-1+\epsilon})$,*

2. *$\theta_2 = O(N^{-\frac{1}{2}+\epsilon})$,*

3. *$\theta_3 = \cdots = \theta_N = 0$.*

Finally, invoking known results on the largest eigenvalues of rank-2 spiked Wigner matrices (e.g., (Renfrew & Soshnikov, 2013)), we can complete the proof of Theorem 2.5.

## 4. Outline of the Proof - Subcritical Case

In this section, we outline the proof of the second part of our main result, Theorem 2.5. (The detailed proofs for the results in Section 3 can be found in Appendix B.) Throughout Section 4, we assume $\lambda_e < 1$.

### 4.1. Approximation by a Wigner-type Matrix

The first step of the proof of Theorem 2.5 is the approximation of the transformed matrix $\widetilde{M}$ by a spiked random matrix $H$ in (12) for which we can directly use Proposition 3.1. In the second step, however, instead of interpolating $H$ and a spiked Wigner matrix $H(0)$ in (17), we directly compare $H$ and a Wigner-type matrix $V$ in (14). We will prove the following result on the difference between the largest eigenvalues of $H$ and $V$.

**Proposition 4.1.** *Let $\mu_1(H)$ and $\mu_1(V)$ be the largest eigenvalues of $H$ and $V$, respectively. Then, $0 \leq \mu_1(H) - \mu_1(V) \leq N^{-3/4}$ with overwhelming probability.*

To prove Proposition 4.1, we compare the normalized trace

of the resolvents of $H$ and $V$. Recall that

$$H_{ij} = \frac{f_{ij}}{\sqrt{N}} + \sqrt{\lambda} f'_{ij} x_i x_j + \frac{\lambda}{2} \mathbb{E}[f'_{ij}] \sqrt{N} x_i^2 x_j^2,$$

$$V_{ij} = \frac{f_{ij}}{\sqrt{N}} + \sqrt{\lambda}(f'_{ij} - \mathbb{E}[f'_{ij}]) x_i x_j.$$

Set

$$G(z) \equiv G(1, z) = (H - zI)^{-1},$$
$$R(z) \equiv R(1, z) = (V - zI)^{-1}.$$

From the resolvent identity on $R - G$, we can prove that $|\operatorname{Tr}(R(z) - G(z))| \ll |\operatorname{Tr} R(z)|$ with high probability. We can then prove Proposition 4.1 by showing that if $\mu_1(H)$ is not sufficiently close to $\mu_1(V)$, then $|\operatorname{Tr} G(z)| \gg |\operatorname{Tr} R(z)|$ for some $z$ close to $\mu_1(H)$ (but not to $\mu_1(V)$). We remark that Proposition 4.1 is not optimal, and the bound can be lowered; however, we do not pursue it in this paper.

### 4.2. Local Law and Green Function Comparison

From Proposition 4.1, we can immediately see that the fluctuation of $\mu_1(H)$ is governed by that of $\mu_1(V)$, which is expected to be of order $N^{-2/3}$. To prove that the fluctuation of $\mu_1(V)$, we estimate the change of the largest eigenvalue along the matrix flow given by $V(t)$, defined in (16). We use the Green function argument, which requires the local law, analogous to Proposition 3.2, for the resolvent of $V(t)$. The precise statement is as follows:

**Proposition 4.2.** *Let $V(t)$ be the matrix defined in (16). Recall that we denote the resolvent of $V(t)$ by $R(t, z)$. Set $z \equiv \tau + i\eta$ and $\kappa := \tau - 2$. Suppose that $|\kappa| \leq N^{-\frac{2}{3}+\epsilon}$ and $\eta = N^{-\frac{2}{3}-\epsilon}$ for a fixed $\epsilon > 0$. Then,*

$$\max_{i,j} |R_{ij}(t, z) - m_{sc}(z)\delta_{ij}| = \mathcal{O}(N^{-\frac{1}{3}+\epsilon}), \qquad (23)$$

*uniformly on $t \in [0, 1]$.*

With the local law in Proposition 4.2, we can prove a statement analogous to Proposition 3.4. By adapting the strategy explained in the beginning of Section 3.3, we can then conclude that the the limiting distribution of the fluctuation of $\mu_1(V)$ is equal to that of the largest eigenvalue of a Wigner matrix $V(0)$, which is given by the Tracy–Widom distribution.

## 5. Numerical Experiments

In this section, we perform numerical experiments compare the theoretical results from Theorem 2.5 with empirical results using specific models.

### 5.1. Spiked Non-Gaussian Wigner Matrix with Entrywise Transformation

We first consider a transformed spiked Wigner matrix with non-Gaussian noise. The (rescaled) noise entries $\sqrt{N}W_{ij}$ ($i \leq j$) are independently drawn from the sum of Gaussian and Rademacher random variables, whose density $p$ is given by a centered bimodal distribution with unit variance. (See Figure 1(a) for the graph of $p$.) For the spike, we sample a random $N$-vector $\boldsymbol{x}$ so that $\sqrt{N}x_i$'s are i.i.d. Rademacher random variables, independent from the noise. We apply the entrywise transformation

$$f = -p'/(\sqrt{F_h}p) \qquad (24)$$

where the Fisher information $F_h$ of $h := -p'/p$ is defined by

$$F_h := \mathbb{E}[h(\sqrt{N}W_{12})^2] = \int_{-\infty}^{\infty} \frac{(p'(x))^2}{p(x)} \, dx.$$

(See Figure 1(b) for the graph of $f$.) We remark that $f$ in (24) is the optimal entrywise transformation in the sense that it maximizes the effective SNR $\lambda_e$, and it also satisfies Assumption 2.4.

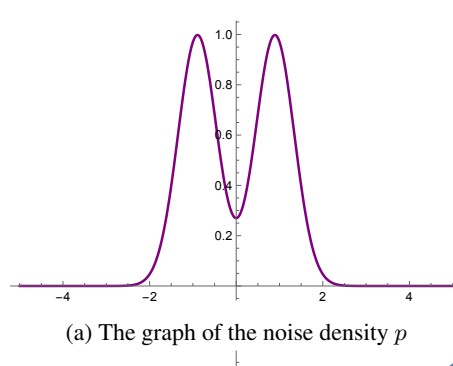

(a) The graph of the noise density $p$

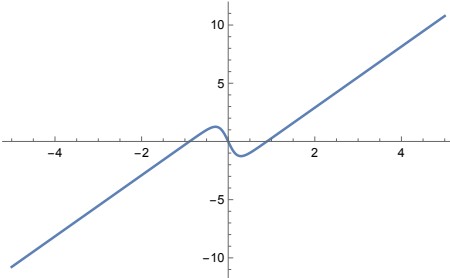

(b) The graph of the entrywise transform $f$

*Figure 1.* The noise density and the entrywise transformation used in the numerical experiment with non-Gaussian bimodal noise.

We set $N = 1024$ and generate 5,000 independent transformed spiked Wigner matrices described above. For a supercritical case, we set SNR $\lambda = 0.8$ with $\lambda_e \approx 2.902$. The histogram for the shifted, rescaled largest eigenvalue $N^{1/2}(\mu_1 - (\sqrt{\lambda_e} + \frac{1}{\sqrt{\lambda_e}}))$ is shown in Figure 2(a). For

a subcritical case, we set SNR $\lambda = 0.1$ with $\lambda_e \approx 0.363$. The histogram for the shifted, rescaled largest eigenvalue $N^{2/3}(\mu_1 - 2)$ is shown in Figure 2(b). For both cases, the empirical distributions closely match the theoretical results. See Appendix A.1 for more detail, including the precise formulas for the noise density $p$ and the entrywise transformation $f$.)

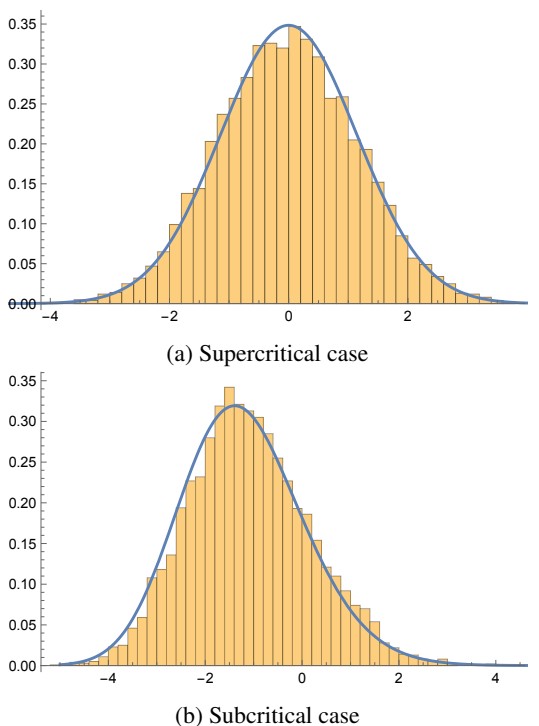

(a) Supercritical case

(b) Subcritical case

*Figure 2.* The sampled distribution of $\mu_1(\widetilde{M})$ for non-Gaussian noise with $N = 1024$, $\lambda = 0.8$(left) and $\lambda = 0.1$(right), after shifting and rescaling. The lines in figure 2(a) plots the Gaussian distribution introduced in Theorem 2.5, and the line in figure 2(b) plots the GOE Tracy–Widom distribution.

## 5.2. Spiked Gaussian Wigner Matrix with Entrywise Transformation

We next consider a spiked Gaussian Wigner matrix, entrywise transformed by a polynomial. We let each noise entry be i.i.d. Gaussian and apply the mapping $f(x) = (x^2 + 3x - 1)/\sqrt{11}$ entrywise. The spike is again sampled using i.i.d. Rademacher random variables.

We set $N = 1024$ and generate 5,000 independent transformed spiked Wigner matrices described above. For a supercritical case, we set SNR $\lambda = 2.5$ with $\lambda_e \approx 1.294$. The histogram for the shifted, rescaled largest eigenvalue $N^{1/2}(\mu_1 - (\sqrt{\lambda_e} + \frac{1}{\sqrt{\lambda_e}}))$ is shown in Figure 3(a). For a subcritical case, we set SNR $\lambda = 0.1$ with $\lambda_e \approx 0.350$. The histogram for the shifted, rescaled largest eigenvalue $N^{2/3}(\mu_1 - 2)$ is shown in Figure 3(b). For both cases, the

empirical distributions closely match the theoretical results. See also Appendix A.2 for additional detail.

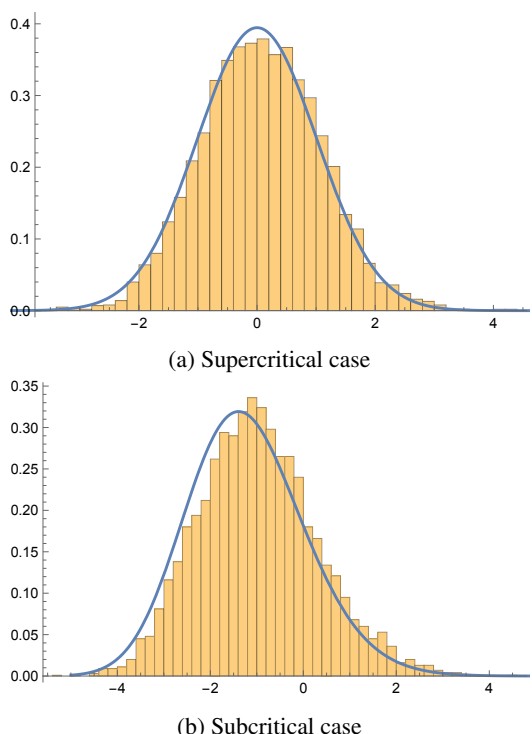

(a) Supercritical case

(b) Subcritical case

*Figure 3.* The sampled distribution of $\mu_1(\widetilde{M})$ for Gaussian noise with $N = 1024$, $\lambda = 2.5$ (left) and $\lambda = 0.15$ (right), after shifting and rescaling. The line in figure 3(a) plots the Gaussian distribution introduced in Theorem 2.5, and the line in figure 3(b) plots the GOE Tracy–Widom distribution.

## 6. Conclusion and Future Works

In this paper, we proved that the largest eigenvalue of the transformed rank-one spiked Wigner matrix exhibits the BBP transition in terms of the fluctuation of the largest eigenvalue. The limiting law of the fluctuation is given by the Gaussian when the effective SNR is above a certain threshold, and by the GOE Tracy–Widom below the threshold. We also proved the precise formulas for the limiting laws involving the effective SNR due to the entrywise transform, and strong concentration estimates for the largest eigenvalues, known as the rigidity. We conducted numerical simulations with several examples to compare the theoretical results with the numerical ones.

A natural future research direction would be to prove the corresponding results for rectangular models, which generalize the spiked Wishart matrices. We believe that our approach would work for the analysis on the largest singular values of transformed spiked rectangular matrices. We also hope to remove several technical assumptions on our model in the future.

## Acknowledgements

The authors would like to thank anonymous reviewers for their suggestions and feedback. This work was partially supported by National Research Foundation of Korea under grant number NRF-2019R1A5A1028324 and RS-2023-NR076695.

## Impact Statement

This paper presents work whose goal is to advance the field of Machine Learning. There are many potential societal consequences of our work, none which we feel must be specifically highlighted here.

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

## A. Details of the Numerical Experiments

In this appendix, we provide the details of the numerical experiments in Section 5.

### A.1. Spiked Non-Gaussian Wigner matrix

The density function $p(x)$ of each noise entry is

$$p(x) = \frac{5}{2\sqrt{2\pi}} \left( e^{-\frac{5}{2}(x-2/\sqrt{5})^2} + e^{-\frac{5}{2}(x+2/\sqrt{5})^2} \right). \tag{25}$$

If we denote by $\mathcal{N}$ and $\mathcal{R}$ a standard Gaussian random variable and a Rademacher random variable, respectively, then $p$ is the density function of a random variable

$$\frac{1}{\sqrt{5}}\mathcal{N} + \frac{2}{\sqrt{5}}\mathcal{R}.$$

which represents the convolution of a rescaled Gaussian density function and a rescaled Rademacher density function.

An optimal entrywise transformation (up to constant multiple) $h(x)$ for the transformed PCA is $h(x) := -\frac{p'(x)}{p(x)}$. For our model, the explicit formula for $h(x)$ is

$$h(x) = \frac{5\left( (x - \frac{2}{\sqrt{5}})e^{-\frac{5}{2}(x-2/\sqrt{5})^2} + (x + \frac{2}{\sqrt{5}})e^{-\frac{5}{2}(x-+2/\sqrt{5})^2} \right)}{e^{-\frac{5}{2}(x-2/\sqrt{5})^2} + e^{-\frac{5}{2}(x+2/\sqrt{5})^2}}. \tag{26}$$

Note that $h$ does not satisfy Assumption 2.4, since $F_h := \mathbb{E}[h(\sqrt{N}W_{12})^2] \approx 3.628$, which is the Fisher information of the noise entry. We rescale $h$ by dividing it by $\sqrt{F_h}$, and thus we are led to apply

$$f(x) := \frac{h(x)}{\sqrt{F_h}} \approx \frac{2.62503\left( (x - \frac{2}{\sqrt{5}})e^{-\frac{5}{2}(x-2/\sqrt{5})^2} + (x + \frac{2}{\sqrt{5}})e^{-\frac{5}{2}(x-+2/\sqrt{5})^2} \right)}{e^{-\frac{5}{2}(x-2/\sqrt{5})^2} + e^{-\frac{5}{2}(x+2/\sqrt{5})^2}}. \tag{27}$$

Note that $\mathbb{E}[f_{12}^2] = 1$ and $\mathbb{E}[f'_{12}] \approx 1.905$. We remark that for the transformed PCA the threshold for the strong detection is $(\mathbb{E}[f'_{12}])^{-2} \approx 0.276$.

### A.2. Spiked Gaussian Wigner Matrix

Recall that we let

$$f(x) = \frac{x^2 + 3x - 1}{\sqrt{11}}.$$

It satisfies Assumption 2.4, since

$$\mathbb{E}[f_{ij}^2] = \frac{m_4}{11} + \frac{7m_2}{11} - \frac{6m_1}{11} + \frac{1}{11} = 1 \tag{28}$$

where $m_\ell$ denotes the $\ell$-th moment of the standard Gaussian distribution. (Note that $m_4 = 3$, $m_2 = 1$, and $m_1 = 0$.) In this setting, $\mathbb{E}[f'_{12}] = \frac{3}{\sqrt{11}} \approx 0.905$, and the threshold for the transformed PCA is $(\mathbb{E}[f'_{12}])^{-2} \approx 1.222$. We remark that the threshold cannot be decreased from 1 when the noise is Gaussian.

## B. Proof of Main Results - Supercritical case

In this appendix, we provide the detail of the proofs of the results for the supercritical case. Throughout Appendix B, we assume that $\lambda_e = \lambda(\mathbb{E}[f'_{12}])^2 > 1$.

In Appendices B and C, we will use the following notions, known as the stochastic domination in random matrix theory, which are useful when making precise estimates that hold with overwhelming probability up to small powers of $N$.

**Definition B.1** (Stochastic domination). Let

$$\xi = \left( \xi^{(N)}(u) \ : \ N \in \mathbb{N}, u \in U^{(N)} \right), \qquad \zeta = \left( \zeta^{(N)}(u) \ : \ N \in \mathbb{N}, u \in U^{(N)} \right)$$

be two families of random variables, where $U^{(N)}$ is a possibly $N$-dependent parameter set. We say that $\xi$ is *stochastically dominated by* $\zeta$ (uniformly in $u$) if for all (small) $\epsilon > 0$ and (large) $D > 0$

$$\sup_{u \in U^{(N)}} \mathbb{P}\left( |\xi^{(N)}(u)| > N^\epsilon \zeta^{(N)}(u) \right) \leq N^{-D}$$

for any sufficiently large $N \geq N_0(\varepsilon, D)$.

We write $\xi \prec \zeta$ or $\xi = \mathcal{O}(\zeta)$, if $\xi$ is stochastically dominated by $\zeta$.

## B.1. Proof of Theorem 2.6

We first prove Proposition 3.1.

*Proof of Proposition 3.1.* From Taylor's expansion,

$$
\begin{aligned}
\widetilde{M}_{ij} &= \frac{f(\sqrt{N}M_{ij})}{\sqrt{N}} = \frac{f(\sqrt{N}W_{ij} + \sqrt{\lambda N}x_i x_j)}{\sqrt{N}} \\
&= \frac{f(\sqrt{N}W_{ij})}{\sqrt{N}} + \sqrt{\lambda}f'(\sqrt{N}W_{ij})x_i x_j + \frac{\lambda}{2}f''(\sqrt{N}W_{ij})\sqrt{N}x_i^2 x_j^2 + O\left( f'''(\sqrt{N}W_{ij})Nx_i^3 x_j^3 \right).
\end{aligned}
\tag{29}
$$

Recall the notation

$$f_{ij} := f(\sqrt{N}W_{ij}), \quad f'_{ij} := f'(\sqrt{N}W_{ij}), \quad f''_{ij} := f''(\sqrt{N}W_{ij}), \quad f_{ij}^{(k)} := f^{(k)}(\sqrt{N}W_{ij}) \quad (k \in \mathbb{Z}^+), \tag{30}$$

defined in (13).

Define $\widetilde{H}$ by

$$\widetilde{H}_{ij} = \frac{f_{ij}}{\sqrt{N}} + \sqrt{\lambda}f'_{ij}x_i x_j + \frac{\lambda\sqrt{N}}{2}f''_{ij}x_i^2 x_j^2.$$

Then, by definition $(\widetilde{M} - \widetilde{H})_{ij} = O\left( f'''_{ij}Nx_i^3 x_j^3 \right)$. For given $\epsilon > 0$, let $\Omega_W^\epsilon$ be the event

$$\Omega_W^\epsilon = \{\max_{i,j} |f'''_{ij}| \leq N^\epsilon\}.$$

Note that $\Omega_W^\epsilon$ holds with overwhelming probability. Thus, the Frobenius norm (Hilbert-Schmidt norm) of $(\widetilde{M} - \widetilde{H})$ is bounded by

$$\|\widetilde{M} - \widetilde{H}\|_F \leq C \left( \sum_{i,j} \left| f'''_{ij}Nx_i^3 x_j^3 \right|^2 \right)^{\frac{1}{2}} \leq C(\max_{i,j}|f'''_{ij}|)N\sum_i |x_i|^6 = \mathcal{O}(N^{-1}).$$

In particular, $\|\widetilde{M} - \widetilde{H}\| = \mathcal{O}(N^{-1})$.

We next consider $(H - \widetilde{H})$. Note that

$$(H - \widetilde{H})_{ij} = \frac{\lambda\sqrt{N}}{2}(f''_{ij} - \mathbb{E}[f''_{ij}])x_i^2 x_j^2.$$

Thus, for any unit vector $v = (v_1, v_2, \dots, v_N) \in \mathbb{R}^N$,

$$\langle v, (H - \widetilde{H})v \rangle = \sum_{i,j} \frac{\lambda\sqrt{N}}{2}(f''_{ij} - \mathbb{E}[f''_{ij}])x_i^2 x_j^2 v_i v_j.$$

If we let $F''$ be a matrix defined by $F''_{ij} = N^{-\frac{1}{2}}(f''_{ij} - \mathbb{E}[f''_{ij}])$, then it is a constant multiple of a Wigner matrix. Since the matrix norm of a Wigner matrix is bounded by a constant with overwhelming probability, by considering the vector whose $i$-th entry is $x_i^2 v_i$, we obtain that

$$\sum_{i,j} \frac{\lambda\sqrt{N}}{2}(f''_{ij} - \mathbb{E}[f''_{ij}])x_i^2 x_j^2 v_i v_j = \mathcal{O}\left( N\sum_i x_i^4 v_i^2 \right) = \mathcal{O}(N^{-1}).$$

Since $v$ was an arbitrary unit vector, we find that $\|H - \widetilde{H}\| = \mathcal{O}(N^{-1})$.

So far, we have shown that $\|\widetilde{M} - H\| = \mathcal{O}(N^{-1})$. Then, applying Weyl's inequality (or the min-max principle for the largest eigenvalue), we conclude that $\mu_1(\widetilde{M}) - \mu_1(H) = \mathcal{O}(N^{-1})$. $\qquad\square$

Next, we prove Theorem 2.6 for the supercritical case by adapting the strategy of the proof of Theorem 4.8 in (Perry et al., 2018).

*Proof of Theorem 2.6 - Supercritical case.* From Proposition 3.1, we find it suffices to prove that

$$\left| \mu_1(H) - \left( \sqrt{\lambda_e} + \frac{1}{\sqrt{\lambda_e}} \right) \right| \prec N^{-\frac{1}{2}}.$$

Let $A$ and $\Delta$ be the matrices defined by

$$A_{ij} = \frac{\sqrt{\lambda}}{N}(f'_{ij} - \mathbb{E}[f'_{ij}]), \qquad \Delta_{ij} = \sqrt{N} A_{ij} x_i x_j.$$

Then, for any unit vector $\boldsymbol{y} = (y_1, \ldots, y_N)$, letting $\boldsymbol{z} = (x_1 y_1, \ldots, x_N y_N)$,

$$\langle \boldsymbol{y}, \Delta \boldsymbol{y} \rangle = \sqrt{N} \sum_{i,j=1}^{N} x_i y_i A_{ij} x_j y_j = \sqrt{N} \langle \boldsymbol{z}, A\boldsymbol{z} \rangle \leq \sqrt{N} \|A\| \cdot \|\boldsymbol{z}\|^2.$$

Since $|x_i| \prec N^{-1/2}$, we have that $\|\boldsymbol{z}\| \prec N^{-1/2} \|\boldsymbol{y}\| = N^{-1/2}$. Further, since $A$ is a constant multiple of a Wigner matrix, we find that $\|A\| \prec 1$. Thus, we have $\langle \boldsymbol{y}, \Delta \boldsymbol{y} \rangle \prec N^{-1/2}$. Since $\boldsymbol{y}$ was arbitrary, it shows that $\|\Delta\| \prec N^{-1/2}$.

We now consider $(H - \Delta)$, whose entries are

$$H_{ij} - \Delta_{ij} = \frac{f_{ij}}{\sqrt{N}} + \sqrt{\lambda}\mathbb{E}[f'_{ij}]x_i x_j + \frac{\lambda}{2}\mathbb{E}[f''_{ij}]\sqrt{N}x_i^2 x_j^2.$$

Note that $(H - \Delta)$ is a rank-2 spiked Wigner matrix. We want to invoke Theorem 2.7 in (Knowles & Yin, 2013), which can be rephrased as follows in our setting.

**Proposition B.2** (Theorem 2.7 in (Knowles & Yin, 2013))**.** *Let* $M = W + \sqrt{\lambda_1}\boldsymbol{x}\boldsymbol{x}^T + \sqrt{\lambda_2}\boldsymbol{y}\boldsymbol{y}^T$, *where* $W$ *is a Wigner matrix and* $\boldsymbol{x} = (x_1, \ldots, x_N), \boldsymbol{y} = (y_1, \ldots, y_N) \in \mathbb{R}$ *with* $\langle \boldsymbol{x}, \boldsymbol{y} \rangle = 0$. *Assume further that* $0 < \lambda_2 < 1 < \lambda_1$ *and* $\mathbb{E}[W_{ii}]^2 = 2N^{-1}$ *for any* $i$. *Then, for the largest eigenvalue* $\mu_1(M)$ *of* $M$,

$$\left| \mu_1(M) - \left( \sqrt{\lambda_1} + \frac{1}{\sqrt{\lambda_1}} \right) \right| \prec N^{-\frac{1}{2}}.$$

Since the variance of the diagonal entries of $(H - \Delta)$ is not $2N^{-1}$ but $N^{-1}$, we introduce a diagonal matrix $D$ with entries

$$D_{ii} = (\sqrt{2} - 1)\frac{f_{ii}}{\sqrt{N}}.$$

From the first part of Definition 2.1 (the definition of the Wigner matrix), we can easily find that $|f_{ii}| \prec 1$. Thus, $\|D\| \prec N^{-1/2}$. Now, if we let $\mu_1(H - \Delta + D)$ be the largest eigenvalue of $(H - \Delta + D)$, then from Proposition B.2, we find that

$$\left| \mu_1(H - \Delta + D) - \left( \sqrt{\lambda_e} + \frac{1}{\sqrt{\lambda_e}} \right) \right| \prec N^{-\frac{1}{2}}.$$

Since

$$|\mu_1(H) - \mu_1(H - \Delta + D)| \leq \|\Delta - D\| \leq \|\Delta\| + \|D\| \prec N^{-\frac{1}{2}}, \tag{31}$$

this concludes the proof of Theorem 2.6 for the supercritical case.

$\qquad\square$

## B.2. Proof of the Local Law and the Green Function Comparison Theorem

The proof of the local law, Proposition 3.2, relies on the following lemma, which is the local law for the resolvent of $V(t)$.

**Lemma B.3.** *Let $V(t)$ be the matrix defined in* (16) *and its resolvent $R(t,z)$ in* (20). *Then,*

$$\max_{i,j}|R_{ij}(t,z) - m_{sc}(z)\delta_{ij}| = \mathcal{O}(N^{-\frac{1}{2}+2\epsilon}), \tag{32}$$

*where we assume that the conditions in Proposition 3.2 hold for $z \equiv \tau + i\eta$.*

*Proof.* Recall the quadratic vector equation (21). In the regime considered in this lemma, we have an estimate for the resolvent $R(t,z)$ that

$$\max_{i,j}|R_{ij}(t,z) - m_i(t,z)\delta_{ij}| = \mathcal{O}(N^{-\frac{1}{2}+\epsilon}). \tag{33}$$

(See Theorem 1.7 in (Ajanki et al., 2017), where the parameters are given as $\kappa(z) = \Theta(1)$ and $\rho(z) = \Theta(\eta)$ from (1.17), (1.23), and (4.5f) in (Ajanki et al., 2017).) For the solution $m_i \equiv m_i(t,z)$ of (21), we can consider an ansatz

$$m_i(t,z) = m_{sc}(z) + s_i(t,z)$$

with $s_i \equiv s_i(t,z) \ll m_{sc}$. We then have from (21) that

$$0 = 1 + zm_i + \sum_j \mathbb{E}[(V(t)_{ij})^2]m_i m_j = 1 + z(m_{sc} + s_i) + \sum_j \mathbb{E}[(V(t)_{ij})^2](m_{sc} + s_i)(m_{sc} + s_j).$$

From (16), applying the identity $1 + zm_{sc} + m_{sc}^2 = 0$, we further have that

$$zs_i - m_{sc}^2 + \frac{1}{N}(m_{sc} + s_i)\sum_j (1 + C_1^V t\sqrt{N}x_i x_j + C_2^V tN x_i^2 x_j^2)(m_{sc} + s_j) = 0,$$

which yields

$$(z + m_{sc})s_i + \frac{m_{sc} + s_i}{N}\sum_j s_j + (m_{sc} + s_i)\frac{C_1^V t}{\sqrt{N}}\sum_j (m_{sc} + s_j)x_i x_j = O(N^{-1+2\epsilon}). \tag{34}$$

Summing the left-hand side of (34) over the index $i$ and dividing it by $N$,

$$(z + 2m_{sc})\left(\frac{1}{N}\sum_i s_i\right) + \left(\frac{1}{N}\sum_i s_i\right)^2 + \frac{C_1^V t}{N\sqrt{N}}\left(\sum_i(m_{sc} + s_i)x_i\right)^2 = O(N^{-1+2\epsilon}). \tag{35}$$

Since $|x_i| = O(N^{-\frac{1}{2}+\epsilon})$, by naive power counting, the last term in the left-hand side of (35) is $O(N^{-\frac{1}{2}+2\epsilon})$. Thus, $\frac{1}{N}\sum_i s_i = O(N^{-\frac{1}{2}+2\epsilon})$. Plugging it into (34), since the last term in the left-hand side of (35) is $O(N^{-\frac{1}{2}+2\epsilon})$, we find that $s_i = O(N^{-\frac{1}{2}+2\epsilon})$ as well. With this bound on $s_i$, we then bound the last term in the left-hand side of (35) by

$$\frac{C_1^V t}{N\sqrt{N}}\left(m_{sc}\sum_i x_i + \sum_i s_i x_i\right)^2 = O(N^{-\frac{3}{2}+6\epsilon}),$$

where we used the assumption that $\sum_i x_i = O(N^\epsilon)$. Thus, from (35), we find that $\frac{1}{N}\sum_i s_i = O(N^{-1+2\epsilon})$, and plugging it again into (34), we conclude that $s_i = O(N^{-1+2\epsilon})$.

Combining the estimate $s_i = O(N^{-1+2\epsilon})$ with the local law (33), we now obtain (32). $\qquad\square$

*Remark B.4.* The local law in (32) can be strengthened further; we will use the averaged local law

$$\left|\frac{1}{N}\sum_{i=1}^N \overline{w}_i(R_{ii}(t,z) - m_i(t,z))\right| = \mathcal{O}(N^{-1+2\epsilon}) \tag{36}$$

for any deterministic vector $(w_1, w_2, \ldots, w_N)$ with $\max_i |w_i| \leq 1$ and the anisotropic local law

$$\left| \sum_{i,j=1}^{N} \overline{w}_i R_{ij}(t,z) v_j - \sum_{i=1}^{N} m_i(t,z) \overline{w}_i v_i \right| = \mathcal{O}(N^{-\frac{1}{2}+2\epsilon}) \tag{37}$$

for any deterministic unit vectors $(w_1, w_2, \ldots, w_N)$ and $(v_1, v_2, \ldots, v_N)$. (See Theorem 1.7 and Theorem 1.13 of (Ajanki et al., 2017), respectively.) With the estimate $s_i = m_i - m_{sc} = O(N^{-1+2\epsilon})$ that we showed in the proof of Lemma B.3, we find that the local laws in (32), (36), and (37) remain valid even if we replace $m_i$ by $m_{sc}$.

With the local laws for $R(t,z)$, we now prove the local law for $G(t,z)$, Proposition 3.2.

*Proof of Proposition 3.2.* We apply the resolvent identity

$$R(t,z) - G(t,z) = R(t,z)(H(t) - V(t))G(t,z), \tag{38}$$

which can be easily checked by multiplying the identity by $(H(t) - zI)$ from the right and by $(V(t) - zI)$ from the left. Let $\boldsymbol{x}^2 := (x_1^2, x_2^2, \ldots, x_N^2)$. Note that $\|\boldsymbol{x}^2\| \leq (\max_i |x_i|)\|\boldsymbol{x}\| = O(N^{-\frac{1}{2}+\epsilon'})$ for any $\epsilon' > 0$. Then,

$$H(t) - V(t) = \sqrt{\lambda}\mathbb{E}[f_{12}']\boldsymbol{x}\boldsymbol{x}^T + \frac{\lambda}{2}\mathbb{E}[f_{12}'']\sqrt{N}\boldsymbol{x}^2(\boldsymbol{x}^2)^T.$$

Set $R \equiv R(t,z)$, $G \equiv G(t,z)$, and let $\boldsymbol{e}_i$ be the $i$-th standard basis vector. Then,

$$R_{ij} - G_{ij} = \langle \boldsymbol{e}_i, (R-G)\boldsymbol{e}_j \rangle = \sqrt{\lambda}\mathbb{E}[f_{12}']\langle \boldsymbol{e}_i, R\boldsymbol{x} \rangle\langle \boldsymbol{x}, G\boldsymbol{e}_j \rangle + \frac{\lambda}{2}\mathbb{E}[f_{12}'']\sqrt{N}\langle \boldsymbol{e}_i, R\boldsymbol{x}^2 \rangle\langle \boldsymbol{x}^2, G\boldsymbol{e}_j \rangle. \tag{39}$$

We want to show that the right-hand side of (39) is negligible. First, by applying the anisotropic local law in (37), we find

$$\langle \boldsymbol{e}_i, R\boldsymbol{x} \rangle = \mathcal{O}(N^{-\frac{1}{2}+2\epsilon}), \quad \langle \boldsymbol{e}_i, R\boldsymbol{x}^2 \rangle = \mathcal{O}(N^{-1+2\epsilon}). \tag{40}$$

To bound the second term in the right-hand side of (39), it suffices to use the trivial estimate $\|G\| \leq \eta^{-1} = N^{\frac{1}{2}+\epsilon}$, which can be readily checked from an inequality $|(\mu_i(H(t)) - z)^{-1}| \leq |\operatorname{Im} z|^{-1} = \eta^{-1}$, where $\mu_i(H(t))$ $(i = 1, 2, \ldots, N)$ are the eigenvalues of $H(t)$. We then find that

$$\frac{\lambda}{2}\mathbb{E}[f_{12}'']\sqrt{N}\langle \boldsymbol{e}_i, R\boldsymbol{x}^2 \rangle\langle \boldsymbol{x}^2, G\boldsymbol{e}_j \rangle = \mathcal{O}(N^{\frac{1}{2}}N^{-1+2\epsilon}\eta^{-1}\|\boldsymbol{x}^2\|) = \mathcal{O}(N^{-\frac{1}{2}+3\epsilon}). \tag{41}$$

We now estimate $\langle \boldsymbol{x}, G\boldsymbol{e}_j \rangle$. From the resolvent identity (38),

$$\langle \boldsymbol{x}, R\boldsymbol{e}_j \rangle - \langle \boldsymbol{x}, G\boldsymbol{e}_j \rangle = \sqrt{\lambda}\mathbb{E}[f_{12}']\langle \boldsymbol{x}, R\boldsymbol{x} \rangle\langle \boldsymbol{x}, G\boldsymbol{e}_j \rangle + \frac{\lambda}{2}\mathbb{E}[f_{12}'']\sqrt{N}\langle \boldsymbol{x}, R\boldsymbol{x}^2 \rangle\langle \boldsymbol{x}^2, G\boldsymbol{e}_j \rangle.$$

Solving it for $\langle \boldsymbol{x}, G\boldsymbol{e}_j \rangle$, we get

$$\langle \boldsymbol{x}, G\boldsymbol{e}_j \rangle = \frac{1}{1 + \sqrt{\lambda}\mathbb{E}[f_{12}']\langle \boldsymbol{x}, R\boldsymbol{x} \rangle} \left( \langle \boldsymbol{x}, R\boldsymbol{e}_j \rangle - \frac{\lambda}{2}\mathbb{E}[f_{12}'']\sqrt{N}\langle \boldsymbol{x}, R\boldsymbol{x}^2 \rangle\langle \boldsymbol{x}^2, G\boldsymbol{e}_j \rangle \right). \tag{42}$$

It should be noted that for our choice of $z$, from the anisotropic local law, $\langle \boldsymbol{x}, R\boldsymbol{x} \rangle$ is near $-1/(\sqrt{\lambda}\mathbb{E}[f_{12}'])$. To obtain an estimate on $\langle \boldsymbol{x}, G\boldsymbol{e}_j \rangle$, we consider the imaginary part of $\langle \boldsymbol{x}, R\boldsymbol{x} \rangle$. From the spectral decomposition,

$$R = \sum_k \frac{\boldsymbol{v}_k(t)\boldsymbol{v}_k(t)^T}{\mu_k(V(t)) - z},$$

where $\boldsymbol{v}_k(t)$ is the normalized eigenvector of $V(t)$ associated with the eigenvalue $\mu_k(V(t))$. Since $V(t)$ is a Wigner-type matrix with $\|V(t) - V(0)\| = o(1)$, it is not hard to see that $\mu_k(V(t)) \geq -3$ for all $k = 1, 2, \ldots, N$ with overwhelming probability. Thus,

$$\operatorname{Im}\langle \boldsymbol{x}, R\boldsymbol{x} \rangle = \operatorname{Im} \sum_k \frac{\boldsymbol{x}^T \boldsymbol{v}_k(t)\boldsymbol{v}_k(t)^T \boldsymbol{x}}{\mu_k(V(t)) - z} = \sum_k |\langle \boldsymbol{x}, \boldsymbol{v}_k(t) \rangle|^2 \operatorname{Im} \frac{1}{\mu_k(V(t)) - z}$$

$$= \sum_k |\langle \boldsymbol{x}, \boldsymbol{v}_k(t) \rangle|^2 \frac{\eta}{(\mu_k(V(t)) - \tau)^2 + \eta^2} \geq \frac{\eta}{(\tau+4)^2} \sum_k |\langle \boldsymbol{x}, \boldsymbol{v}_k(t) \rangle|^2 = \frac{\eta}{(\tau+4)^2}$$

with high probability, since $\{\boldsymbol{v}_k(V(t))\}$ forms an eigenbasis. We now have that

$$|1 + \sqrt{\lambda}\mathbb{E}[f'_{12}]\langle \boldsymbol{x}, R\boldsymbol{x}\rangle| \geq \sqrt{\lambda}\mathbb{E}[f'_{12}]\operatorname{Im}\langle \boldsymbol{x}, R\boldsymbol{x}\rangle \geq C\eta \tag{43}$$

for some constant $C$ with high probability. Furthermore, from the anisotropic local law in (37), we find that

$$\langle \boldsymbol{x}, R\boldsymbol{x}^2\rangle = \sum_{i=1}^{N} m_i x_i^3 + \mathcal{O}(N^{-1+2\epsilon}) = m_{sc}\sum_{i=1}^{N} x_i^3 + \sum_{i=1}^{N}(m_i - m_{sc})x_i^3 + \mathcal{O}(N^{-1+2\epsilon})$$
$$= \mathcal{O}(N^{-1+\epsilon'} + N^{-\frac{3}{2}+5\epsilon} + N^{-1+2\epsilon}).$$

Thus, following the argument we used to derive (41), we obtain

$$\frac{\lambda}{2}\mathbb{E}[f''_{12}]\sqrt{N}\langle \boldsymbol{x}, R\boldsymbol{x}^2\rangle\langle \boldsymbol{x}^2, G\boldsymbol{e}_j\rangle = \mathcal{O}(N^{-\frac{1}{2}+3\epsilon}). \tag{44}$$

Plugging (43) and (44) into (42), together with (40), we get

$$\langle \boldsymbol{x}, G\boldsymbol{e}_j\rangle = \mathcal{O}(\eta^{-1}N^{-\frac{1}{2}+3\epsilon}) = \mathcal{O}(N^{4\epsilon}).$$

Finally, going back to the resolvent identity (39), from (32), (40), and (41), we conclude that (19) holds. This completes the proof of Proposition 3.2. □

Next, we prove the Green function comparison theorem, Proposition 3.4.

*Proof of Proposition 3.4.* Fix $x \in [E_1, E_2]$. For simplicity, set

$$G \equiv G(t, x + \mathrm{i}\eta), \quad X \equiv \operatorname{Im}\int_{E_1}^{E_2}\operatorname{Tr}G(t, x + \mathrm{i}\eta)\mathrm{d}x.$$

From our choice of $x$ and $\eta$, and the assumption on $F$, we have from the local law for $G$, Proposition 3.2, that

$$X = \mathcal{O}(N^{C\epsilon}), \quad F^{(m)}(X) = \mathcal{O}(N^{C\epsilon})$$

for some constant $C$. (Throughout the proof, we use $C$ to denote positive constants independent of $N$, whose value may change from line to line.) Differentiating $F(X)$ with respect to $t$,

$$\frac{\mathrm{d}}{\mathrm{d}t}\mathbb{E}F(X) = \mathbb{E}\left[F'(X)\frac{\mathrm{d}X}{\mathrm{d}t}\right] = \mathbb{E}\left[F'(X)\operatorname{Im}\int_{E_1}^{E_2}\sum_i \frac{\mathrm{d}G_{ii}}{\mathrm{d}t}\mathrm{d}x\right]$$
$$= \mathbb{E}\left[F'(X)\operatorname{Im}\int_{E_1}^{E_2}\sum_i\sum_{j\leq k}\frac{\mathrm{d}H(t)_{jk}}{\mathrm{d}t}\frac{\partial G_{ii}}{\partial H(t)_{jk}}\mathrm{d}x\right]. \tag{45}$$

From the definition of $H(t)$ in (17) and (16),

$$\dot{H}(t)_{jk} := \frac{\mathrm{d}H(t)_{jk}}{\mathrm{d}t} = \frac{C_1^V\sqrt{N}x_jx_k + C_2^V Nx_j^2x_k^2}{2(1 + C_1^V t\sqrt{N}x_jx_k + C_2^V tNx_j^2x_k^2)^{1/2}}\frac{V_{jk}}{\sqrt{N\mathbb{E}[(V_{jk})^2]}}$$
$$= \frac{C_1^V\sqrt{N}x_jx_k + C_2^V Nx_j^2x_k^2}{2(1 + C_1^V t\sqrt{N}x_jx_k + C_2^V tNx_j^2x_k^2)}V(t)_{jk}.$$

Applying the well-known formula for the derivative of the resolvent,

$$\frac{\partial G_{ab}}{\partial H(t)_{jk}} = \begin{cases} -G_{aj}G_{kb} - G_{ak}Gjb & \text{if } j \neq k \\ -G_{aj}Gjb & \text{if } j = k \end{cases},$$

we find that

$$\frac{\mathrm{d}}{\mathrm{d}t}\mathbb{E}F(X) = -\operatorname{Im}\int_{E_1}^{E_2}\mathbb{E}\left[F'(X)\sum_{i,j,k}\frac{C_1^V\sqrt{N}x_jx_k + C_2^V Nx_j^2x_k^2}{2(1+C_1^V t\sqrt{N}x_jx_k + C_2^V tNx_j^2x_k^2)}V(t)_{jk}G_{ij}G_{ki}\right]\mathrm{d}x. \tag{46}$$

To prove that the derivative of $\mathbb{E}[F(X)]$ is small, we use Stein's method to bound the right-hand side of (46). Notice that

$$\frac{\partial G_{ab}}{\partial V(t)_{jk}} = \frac{\partial G_{ab}}{\partial H(t)_{jk}},$$

which can be readily checked from the definition of $H(t)$ in (17). Let

$$\Omega_\epsilon = \{\max_{i,j}|G_{ij} - m_{sc}\delta_{ij}| < N^{-\frac{1}{2}+7\epsilon} \text{ for all } x \in [E_1, E_2]\}.$$

Then, $\Omega_\epsilon$ holds with high probability. On $\Omega_\epsilon$, when we expand the right-hand side of (46), the terms involving the third or higher cumulants are negligible in the sense that it is $O(N^{-\frac{1}{2}+C\epsilon})$. Indeed, if we perform the naive power counting, we get a factor $N^{-\frac{3}{2}}$ from the third (or higher) cumulant, $N^{-\frac{1}{2}+\epsilon}$ from the length $|E_2 - E_1|$ of the interval in the integral, $N^3$ from the three summation indices $i, j, k$, $N^{-\frac{1}{2}+2\epsilon}$ from the factor $(C_1^V\sqrt{N}x_jx_k + C_2^V Nx_j^2x_k^2)$, and $N^{-1+C\epsilon}$ since each term contains at least two off-diagonal entries of $G$, hence the total size of these terms are at most $\mathcal{O}(N^{-\frac{3}{2}}N^{-\frac{1}{2}+\epsilon}N^3 N^{-\frac{1}{2}+2\epsilon}N^{-1+C\epsilon}) = \mathcal{O}(N^{-\frac{1}{2}+C\epsilon})$. (While some terms may contain less than two off-diagonal entries in case some summation indices coincide, the power from the summation indices decreases to $N^2$ in such cases.)

From the power counting argument above, on $\Omega_\epsilon$, we find that

$$\frac{\mathrm{d}}{\mathrm{d}t}\mathbb{E}F(X)$$

$$= -\operatorname{Im}\int_{E_1}^{E_2}\sum_{i,j,k}\mathbb{E}[(V(t)_{jk})^2]\frac{C_1^V\sqrt{N}x_jx_k + C_2^V Nx_j^2x_k^2}{2(1+C_1^V t\sqrt{N}x_ix_j + C_2^V tNx_i^2x_j^2)}\mathbb{E}\left[\frac{\partial}{\partial V(t)_{jk}}(F'(X)G_{ij}G_{ki})\right]\mathrm{d}x$$

$$+ \mathcal{O}(N^{-\frac{1}{2}+C\epsilon})$$

$$= -\frac{1}{2N}\operatorname{Im}\int_{E_1}^{E_2}\sum_{i,j,k}(C_1^V\sqrt{N}x_jx_k + C_2^V Nx_j^2x_k^2)\mathbb{E}\left[F''(X)\frac{\partial X}{\partial V(t)_{jk}}G_{ij}G_{ki} + F'(X)\frac{\partial(G_{ij}G_{ki})}{\partial V(t)_{jk}}\right]\mathrm{d}x$$

$$+ \mathcal{O}(N^{-\frac{1}{2}+C\epsilon}).$$

(In the rest of the proof, we always assume that $\Omega_\epsilon$ holds.) We notice that,

$$\frac{\partial X}{\partial V(t)_{jk}} = -\operatorname{Im}\int_{E_1}^{E_2}\sum_a(G_{aj}G_{ka} + G_{ak}G_{ja})\mathrm{d}x = O(N^{-\frac{1}{2}+C\epsilon})$$

from the power counting; here the factors are $N^{-\frac{1}{2}+\epsilon}$ from the length $|E_2 - E_1|$, $N$ from the summation index, and $N^{-1+C\epsilon}$ from two off-diagonal entries of $G$. (The notation above illustrates the case where $j \neq k$; however, we easily see that the same calculation applies for $j = k$.) Then, again by power counting,

$$\frac{1}{N}\int_{E_1}^{E_2}\sum_{i,j,k}(C_1^V\sqrt{N}x_jx_k + C_2^V Nx_j^2x_k^2)\mathbb{E}\left[F''(X)\frac{\partial X}{\partial V(t)_{jk}}G_{ij}G_{ki}\right]\mathrm{d}x = O(N^{-\frac{1}{2}+C\epsilon}).$$

It remains to bound

$$-\frac{1}{2N}\int_{E_1}^{E_2}\sum_{i,j,k}(C_1^V\sqrt{N}x_jx_k + C_2^V Nx_j^2x_k^2)\mathbb{E}\left[F'(X)\frac{\partial(G_{ij}G_{ki})}{\partial V(t)_{jk}}\right]\mathrm{d}x$$

$$= \frac{1}{N}\int_{E_1}^{E_2}\sum_{i,j,k}(C_1^V\sqrt{N}x_jx_k + C_2^V Nx_j^2x_k^2)\mathbb{E}\left[F'(X)(G_{ij}G_{kk}G_{ji} + G_{ij}G_{kj}G_{ki})\right]\mathrm{d}x. \tag{47}$$

For the term with three off-diagonal entries of $G$, we can again use the power counting to show that it is $O(N^{-\frac{1}{2}+C\epsilon})$. Similarly, with the factor $Nx_j^2 x_k^2 = O(N^{-1+4\epsilon})$, we can find that

$$\frac{1}{N}\int_{E_1}^{E_2}\sum_{i,j,k}C_2^V Nx_j^2 x_k^2\mathbb{E}\left[F'(X)G_{ij}G_{kk}G_{ji}\right]\mathrm{d}x = O(N^{-\frac{1}{2}+C\epsilon}).$$

For the remaining term,

$$\frac{1}{N}\int_{E_1}^{E_2}\sum_{i,j,k}C_1^V\sqrt{N}x_j x_k\mathbb{E}\left[F'(X)G_{ij}G_{kk}G_{ji}\right]\mathrm{d}x$$

$$= \frac{1}{N}\int_{E_1}^{E_2}\sum_{i,j,k}C_1^V\sqrt{N}x_j x_k\mathbb{E}\left[F'(X)G_{ij}m_{sc}G_{ji}\right]\mathrm{d}x \tag{48}$$

$$+ \frac{1}{N}\int_{E_1}^{E_2}\sum_{i,j,k}C_1^V\sqrt{N}x_j x_k\mathbb{E}\left[F'(X)G_{ij}(G_{kk}-m_{sc})G_{ji}\right]\mathrm{d}x.$$

The second term in the right-hand side of (48) is $O(N^{-\frac{1}{2}+C\epsilon})$ again by power counting, where we use the local law $|G_{kk}-m_{sc}| = O(N^{-\frac{1}{2}+C\epsilon})$, proved in Proposition 3.2. The first term in the right-hand side of (48) can be bounded as

$$\frac{1}{N}\int_{E_1}^{E_2}\sum_{i,j,k}C_1^V\sqrt{N}x_j x_k\mathbb{E}\left[F'(X)G_{ij}m_{sc}G_{ji}\right]\mathrm{d}x$$

$$= \frac{C_1^V m_{sc}}{\sqrt{N}}\int_{E_1}^{E_2}\left(\sum_k x_k\right)\sum_{i,j}x_j\mathbb{E}\left[F'(X)G_{ij}G_{ji}\right]\mathrm{d}x = O(N^{-\frac{1}{2}+C\epsilon}),$$

where we use the assumption that $\sum_k x_k = O(N^\epsilon)$.

So far, we have seen that $\frac{\mathrm{d}}{\mathrm{d}t}\mathbb{E}F(X) = O(N^{-\frac{1}{2}+C\epsilon})$ for any sufficiently small $\epsilon > 0$. Integrating it from $t=0$ to $t=1$, we can conclude that (22) holds. This completes the proof of Proposition 3.4. $\qquad\square$

### B.3. Proof of the Main Theorem - Supercritical case

We begin by proving Proposition 3.5.

*Proof of Proposition 3.5.* It is obvious that $\theta_3 = \cdots = \theta_N = 0$ since $A_N$ is a rank-2 matrix. To ease the notation, we assume that $\mathbb{E}[f_{12}''] \geq 0$ and denote $A_N = \boldsymbol{u}\boldsymbol{u}^T + \boldsymbol{v}\boldsymbol{v}^T$, where

$$\boldsymbol{u} = (\sqrt{\lambda}\mathbb{E}[f_{12}'])^{1/2}\boldsymbol{x}$$
$$\boldsymbol{v} = (\frac{\lambda}{2}\mathbb{E}[f_{12}'']\sqrt{N})^{1/2}\boldsymbol{x}^2. \tag{49}$$

(The case $\mathbb{E}[f_{12}''] < 0$ can be proved in a similar manner.) It can be readily shown that the two nontrivial eigenvalues of $A_N$ are given by

$$\frac{\|\boldsymbol{u}\|^2 + \|\boldsymbol{v}\|^2 \pm \sqrt{(\|\boldsymbol{u}\|^2 - \|\boldsymbol{v}\|^2)^2 + 4\langle\boldsymbol{u},\boldsymbol{v}\rangle^2}}{2}.$$

By substituting $\boldsymbol{u}$ and $\boldsymbol{v}$, we get

$$\theta_1 = \frac{\sqrt{\lambda}\mathbb{E}[f_{12}'] + \frac{\lambda}{2}\mathbb{E}[f_{12}'']\sqrt{N}(\sum_i x_i^4) + \sqrt{\lambda\mathbb{E}[f_{12}']^2 + \Theta(N,\boldsymbol{x})}}{2} \tag{50}$$

and

$$\theta_2 = \frac{\sqrt{\lambda}\mathbb{E}[f_{12}'] + \frac{\lambda}{2}\mathbb{E}[f_{12}'']\sqrt{N}(\sum_i x_i^4) - \sqrt{\lambda\mathbb{E}[f_{12}']^2 + \Theta(N,\boldsymbol{x})}}{2} \tag{51}$$

where we define

$$\Theta(N, \boldsymbol{x}) := \frac{\lambda^2}{4}\mathbb{E}[f_{12}'']^2 N(\sum_i x_i^4)^2 + \lambda\sqrt{\lambda}\mathbb{E}[f_{12}']\mathbb{E}[f_{12}'']\sqrt{N}(2(\sum_i x_i^3)^2 - \sum_i x_i^4).$$

Note that for any $\epsilon > 0$,

$$\Theta(N, \boldsymbol{x}) = -2\left(\sqrt{\lambda}\mathbb{E}[f_{12}']\right)\left(\frac{\lambda}{2}\mathbb{E}[f_{12}'']\sqrt{N}(\sum_i x_i^4)\right) + O(N^{-1+\epsilon}). \tag{52}$$

Then,

$$\sqrt{\lambda\mathbb{E}[f_{12}']^2 + \Theta(N, \boldsymbol{x})} = \sqrt{\lambda}\mathbb{E}[f_{12}']\left(1 + \frac{\Theta(N, \boldsymbol{x})}{2(\sqrt{\lambda}\mathbb{E}[f_{12}'])^2} + O(N^{-1+\epsilon})\right)$$

$$= \sqrt{\lambda}\mathbb{E}[f_{12}'] - \frac{\lambda}{2}\mathbb{E}[f_{12}'']\sqrt{N}(\sum_i x_i^4) + O(N^{-1+\epsilon}), \tag{53}$$

and plugging it into (50), we get

$$\theta_1 = \sqrt{\lambda}\mathbb{E}[f_{12}'] + O(N^{-1+\epsilon}) \tag{54}$$

and

$$\theta_2 = \frac{\lambda}{2}\mathbb{E}[f_{12}'']\sqrt{N}(\sum_i x_i^4) + O(N^{-1+\epsilon}) = O(N^{-\frac{1}{2}+\epsilon}). \tag{55}$$

This proves the desired proposition. $\qquad\square$

The proof of Proposition 3.3 essentially follows the same strategy as the proof of Proposition 7.1 in (Lee & Schnelli, 2018). However, since the asymptotic spectrum is determined thanks to Theorem 2.1 of (Capitaine et al., 2009), we take a more straightforward approach as follows.

*Proof of Proposition 3.3.* Let $\mu_k(t) \equiv \mu_k(H(t))$ be the $k$-th largest eigenvalues of $H(t)$. Recall that in the proof of Theorem 2.6 in Appendix B.1, we have seen that $\|H - (H - \Delta + D)\| \prec N^{-1/2}$ and $(H - \Delta + D)$ is a rank-2 spiked Wigner matrix. By Proposition 3.5, Theorem 2.1 in (Capitaine et al., 2009), and Theorem 2.7 (Knowles & Yin, 2013), with overwhelming probability,

$$|\mu_k(t) - L| > \frac{L - 2}{2}$$

for $k = 2, 3, \ldots, N$. To prove the proposition, we compare $(\chi_E * \theta_{\eta_2})(\lambda_i)$ and $\chi_{E-\eta_1}$ for the following cases.

*Case* 1. For $x < \frac{L+2}{2}$, $\chi_{E-\eta_1}(x) = 0$. Then we get

$$(\chi_E * \theta_{\eta_2})(x) = \frac{1}{\pi}\left(\tan^{-1}\frac{E_+ - x}{\eta_2} - \tan^{-1}\frac{E - x}{\eta_2}\right) = \frac{1}{\pi}\left(\tan^{-1}\frac{\eta_2}{E - x} - \tan^{-1}\frac{\eta_2}{E_+ - x}\right)$$

$$< \frac{1}{2}\left(\frac{\eta_2}{E - x} - \frac{\eta_2}{E_+ - x}\right) = \frac{1}{2}\frac{\eta_2(E_+ - E)}{(E - x)(E_+ - x)} < N^{-1-\frac{3}{2}\epsilon}. \tag{56}$$

Therefore,

$$\sum_{i:\lambda_i < E - \frac{L-2}{2}}((\chi_E * \theta_{\eta_2})(\lambda_i) - \chi_{E-\eta_1}(\lambda_i)) \le N \cdot 2N^{-1-\frac{3}{2}\epsilon} \le N^{-\epsilon} \tag{57}$$

Observe that only one eigenvalue of $H(t)$, namely $\mu_1(t)$, can be in the following two cases with overwhelming probability.

*Case* 2. For $x \in [\frac{L+2}{2}, E - \eta_1)$, $\chi_{E-\eta_1}(x) = 0$. Similarly to the formula shown above,

$$(\chi_E * \theta_{\eta_2})(x) = \frac{1}{\pi}\left(\tan^{-1}\frac{E_+ - x}{\eta_2} - \tan^{-1}\frac{E - x}{\eta_2}\right) = \frac{1}{\pi}\left(\tan^{-1}\frac{\eta_2}{E - x} - \tan^{-1}\frac{\eta_2}{E_+ - x}\right)$$

$$< \frac{1}{2}\left(\frac{\eta_2}{E - x} - \frac{\eta_2}{E_+ - x}\right) = \frac{1}{2}\frac{\eta_2(E_+ - E)}{(E - x)(E_+ - x)} \le N^{-\epsilon} \tag{58}$$

Therefore,

$$\sum_{i:E-\frac{L-2}{2}\leq\lambda_i<E-\eta_1}((\chi_E*\theta_{\eta_2})(\lambda_i)-\chi_{E-\eta_1}(\lambda_i))\leq N^{-\epsilon} \tag{59}$$

*Case* 3. For $x\in[E-\eta_1,E_+)$, we use the trivial estimate

$$(\chi_E*\theta_{\eta_2})(x)<1=\chi_{E-\eta_1}(x). \tag{60}$$

*Case* 4. There are no eigenvalues in $[E_+,\infty)$ with overwhelming probability.

Considering all cases above, with overwhelming probability, we observe that

$$\mathrm{Tr}(\chi_E*\theta_{\eta_2})(H)\leq\mathrm{Tr}\,\chi_{E-\eta_1}(H)+N^{-\epsilon}. \tag{61}$$

Since $\mathrm{Tr}\,\chi_{E-\eta_1}(H)$ is an integer, by the definition of the cutoff $K$,

$$K(\mathrm{Tr}\,\chi_{E-\eta_1}(H)+N^{-\epsilon})=K(\mathcal{N}_{[E-\eta_1,E_+]}),$$

where $\mathcal{N}_{[E-\eta_1,E_+]}:=|\{i:\lambda_i\in[E-\eta_1,E_+]\}|$. Since $K$ is monotone decreasing on $[0,\infty)$, (61) implies that

$$K(\mathrm{Tr}(\chi_E*\theta_2)(H))\geq K(\mathcal{N}_{[E-\eta_1,E_+]})$$

with overwhelming probability. By taking expectation on both sides, we find that

$$\mathbb{E}[K(\mathrm{Tr}(\chi_E*\theta_2)(H))]>\mathbb{P}(\lambda_1\leq E-\eta_1)-N^{-D},$$

for any $D>0$. The second part of the proposition can also be proved using similar approach by showing that

$$\mathrm{Tr}(\chi_E*\theta_{\eta_2})(H)\geq\mathrm{Tr}\,\chi_{E+\eta_1}(H)-N^{-D}.$$

$\square$

To finish the proof of Theorem 2.5, we use the following result on the fluctuation of the outlier eigenvalues of spiked Wigner matrices. Let

$$c_{\theta_j}:=\frac{\theta_j^2}{\theta_j^2-1},\quad\rho_j=\theta_j+\frac{1}{\theta_j}.$$

**Proposition B.5** (Theorem 1.3 in (Renfrew & Soshnikov, 2013)). *Let $\lambda_1\geq\cdots\geq\lambda_N$ be ordered eigenvalues of $W_N+A_N$. Let $\boldsymbol{u}_N^1,\ldots,\boldsymbol{u}_N^{k_j}$ be a set of orthogonal eigenvectors of $A_N$ with eigenvalue $\theta_j>1$. When the eigenvectors of $A_N$ are delocalized, the difference between the $k_j-$dimensional vector*

$$(c_{\theta_j}\sqrt{N}(\lambda_{k_1+\cdots+k_{j-1}+i}-\rho_j),\,i=1,\ldots,k_j.)$$

*and the vector formed by the (ordered) eigenvalues of a $k_j\times k_j$ GOE matrix with the variance of the matrix entries given by $\frac{\theta_j^2}{\theta_j^2-1}$ plus a deterministic matrix with lpth entry $(1\leq l,p\leq k_j)$ given by $\frac{1}{\theta_j^2 N}(\boldsymbol{u}_N^l)^*M_3\boldsymbol{u}_N^p$ converged to zero in probability with*

$$(M_3)_{ij}:=\mathbb{E}[(\sqrt{N}W_{ij})^3](1-\delta_{ij}).$$

We now prove the first part of Theorem 2.5.

*Proof of Theorem 2.5 - Supercritical case.* Fix $\epsilon>0$ and let $\eta_1:=N^{-1/2-\epsilon/2}$ and $\eta_2:=N^{-1/2-3\epsilon}$. Consider $E=L+sN^{-1/2}$ with $s\in(-N^\epsilon,N^\epsilon)$ and $E_+:=L+2N^{-1/2+\epsilon}$. Recall that $\mu_k(t)\equiv\mu_k(H(t))$ is the $k$-th largest eigenvalues of $H(t)$. By Proposition 3.3, we get

$$\mathbb{P}(\mu_1(1)\leq E)<\mathbb{E}K(\mathrm{Tr}(\chi_{E+\eta_1}*\theta_{\eta_2})(H))+N^{-D}$$

for any $D > 0$. By applying Proposition 3.4 with $3\epsilon$ instead of $\epsilon$, we find that

$$
\mathbb{E}K(\mathrm{Tr}(\chi_{E+\eta_1} * \theta_{\eta_2})(H)) = \mathbb{E}K\left(\mathrm{Im}\,\frac{N}{\pi}\int_{E+\eta_1}^{E_+}\mathrm{Tr}\,G(1, x + \mathrm{i}\eta_2)\,dx\right)
$$

$$
\leq \mathbb{E}K\left(\mathrm{Im}\,\frac{N}{\pi}\int_{E+\eta_1}^{E_+}\mathrm{Tr}\,G(0, x + \mathrm{i}\eta_2)\,dx\right) + N^{-\delta} = \mathbb{E}K\left(\mathrm{Tr}(\chi_{E+\eta_1} * \theta_{\eta_2})(H(0))\right) + N^{-\delta}
$$

$$(62)$$

for some $\delta > 0$. We can apply Proposition 3.3 again to $H(0)$. Consequently, we obtain

$$
\mathbb{P}\left(N^{1/2}(\mu_1(1) - L) \leq s\right) = \mathbb{P}(\mu_1(1) \leq E)
$$

$$
< \mathbb{E}K\left(\mathrm{Tr}(\chi_{E+\eta_1} * \theta_{\eta_2})(H(1))\right) + N^{-D} < \mathbb{E}K\left(\mathrm{Tr}(\chi_{E+\eta_1} * \theta_{\eta_2})(H(0))\right) + N^{-D} + N^{-\delta}
$$

$$
< \mathbb{P}(\mu_1(0) \leq E) + 2N^{-D} + N^{-\delta} = \mathbb{P}\left(N^{1/2}(\mu_1(0) - L) \leq s\right) + 2N^{-D} + N^{-\delta}
$$

$$(63)$$

Similarly, we also find that

$$
\mathbb{P}\left(N^{1/2}(\mu_1(1) - L) \leq s\right) > \mathbb{P}\left(N^{1/2}(\mu_1(0) - L) \leq s\right) - 2N^{-D} - N^{-\delta} \tag{64}
$$

Since the fluctuation of $\mu_1(0)$ coincides with that of $\mu_1(\widetilde{M})$ from Proposition 3.1, it remains to prove the limiting distribution of $N^{1/2}(\mu_1(0))$. We apply Proposition B.5 to the largest eigenvalue $\mu_1(0)$ of $H(0) = V(0) + A_N$ to find that the fluctuation of

$$
N^{1/2}\left(\mu_1(0) - (\theta_1 + \frac{1}{\theta_1})\right)
$$

coincides with that of a Gaussian random variable with variance $2\theta_1^2/(\theta_1^2 - 1)$. Further, from Proposition 3.5, we can replace $\theta_1$ by $\sqrt{\lambda_e}$ with negligible error. To find the mean of the limiting Gaussian, we need to compute the deterministic term involving $M_3$ in Proposition B.5. For simplicity, we let

$$
\gamma_3 := \mathbb{E}[(V(0)_{12})^3]
$$

so $(\mathrm{M}_3)_{ij} = \gamma_3(1 - \delta_{ij})$ since $V(0)$ is a real Wigner matrix. Then,

$$
\frac{1}{\theta^2 N}\boldsymbol{x}^T\mathrm{M}_3\boldsymbol{x} = \frac{1}{\theta^2 N}\left((\sum_i x_i)^2 - \gamma_3\right) = O(N^{-1+2\epsilon}) \tag{65}
$$

with the assumption $\sum_i x_i = O(N^\epsilon)$. We thus find that the mean of the limiting Gaussian is 0, which concludes the proof of Theorem 2.5. $\qquad\square$

## C. Proof of Main Results - Subcritical Case

In this appendix, we provide the detail of the proofs of the results for the subcritical case. Throughout this appendix we assume that $\lambda_e = \lambda(\mathbb{E}[f'_{12}])^2 < 1$.

### C.1. Proof of the Local Law and Theorem 2.6

We begin by proving the local law, Proposition 4.2.

*Proof of Proposition 4.2.* Recall that we have seen in the proof of B.3 that $|m_i(z) - m_{sc}(z)| \prec O(N^{-1})$. Since this estimate does not depend on $z$,

$$
\max_{i,j}|R_{ij}(t, z) - m_i(t, z)\delta_{ij}| = \mathcal{O}(N^{-\frac{1}{3}+\epsilon}),
$$

which was to be proved. (See Theorem 1.7 in (Ajanki et al., 2017), where the parameters are given as $\kappa(z) = \Theta(1)$ and $\rho(z) = \Theta(\sqrt{\kappa + \eta})$ from (1.17), (1.23), and (4.5d) in (Ajanki et al., 2017).) $\qquad\square$

We first notice that Proposition 3.1 holds for the subcritical case without any change. Thus, Theorem 2.6 is a simple corollary of Proposition 4.1, since from the rigidity of the eigenvalues of Wigner-type matrices, $|\mu_1(V) - 2| \prec N^{-2/3}$. (We refer to Corollary 1.11 in (Ajanki et al., 2017) for more detail.) We now prove Proposition 4.1.

*Proof of Proposition 4.1.* The first inequality of the proposition, $\mu_1(H) \geq \mu_1(V)$, is obvious from the min-max principle for the largest eigenvalue. We now suppose that $\mu_1(H) - \mu_1(V) > N^{-3/4}$.

Fix $\epsilon > 0$. We choose the parameter $z \equiv \tau + i\eta$ with $\tau = \mu_1(V) + \xi$ for $\xi \in [N^{-3/4}, N^{-1/2+\epsilon}]$ and $\eta = N^{-5/6}$. Set $R \equiv R(z) = (V - zI)^{-1}$, $G \equiv G(z) = (H - zI)^{-1}$. Our first goal is to show that $|R_{ii} - G_{ii}| = \mathcal{O}(N^{-1/3})$. To prove this, we start with the resolvent identity

$$R_{ii} - G_{ii} = \langle e_i, (R - G)e_i \rangle = \sqrt{\lambda}\mathbb{E}[f'_{12}]\langle e_i, Rx \rangle\langle x, Ge_i \rangle + \frac{\lambda}{2}\mathbb{E}[f''_{12}]\sqrt{N}\langle e_i, Rx^2 \rangle\langle x^2, Ge_i \rangle, \tag{66}$$

which is obtained by letting $i = j$ in (39). To estimate the terms involving $R$ in (66), we apply the anisotropic local law, Theorem 1.13 in (Ajanki et al., 2017), which is

$$|\langle w, Rv \rangle - m_{sc}(z)\langle w, v \rangle| \prec N^{-1/6} \tag{67}$$

for any unit vectors $w, v$. (Here, the parameters are given as $\kappa(z) = \Theta(1)$ and $\rho(z) = O(\eta) = O(N^{-5/6})$ from (1.17), (1.23), and (4.5d) in (Ajanki et al., 2017).) From (67), we find that

$$\langle x, Re_i \rangle = \mathcal{O}(N^{-1/6}), \quad \langle x, Rx^2 \rangle = \mathcal{O}(N^{-2/3}).$$

To prove an estimate for $\langle x, Ge_i \rangle$, we use a bootstrap argument. In (42) in the proof of Proposition 3.2, by using the resolvent identity in (38), we had

$$\langle x, Ge_i \rangle = \frac{1}{1 + \sqrt{\lambda}\mathbb{E}[f'_{12}]\langle x, Rx \rangle} \left( \langle x, Re_i \rangle - \frac{\lambda}{2}\mathbb{E}[f''_{12}]\sqrt{N}\langle x, Rx^2 \rangle\langle x^2, Ge_i \rangle \right). \tag{68}$$

From (67), $\langle x, Rx \rangle = -1 + o(1)$ with overwhelming probability. Since $\sqrt{\lambda}\mathbb{E}[f'_{12}] < 1$, we find that

$$|1 + \sqrt{\lambda}\mathbb{E}[f'_{12}]\langle x, Rx \rangle| > c > 0$$

with overwhelming probability, for some ($N$-independent) constant $c$. Then, applying the trivial bound $\|G\| \leq \eta^{-1} = N^{5/6}$, from (68), we find $\langle x, Ge_i \rangle = \mathcal{O}(N^{1/6})$, which serves as an a priori estimate. To improve this bound, we consider

$$\langle x^2, Re_i \rangle - \langle x^2, Ge_i \rangle$$

using the resolvent identity in (38) and solve it for $\langle x^2, Ge_i \rangle$. Then, we get

$$\langle x^2, Ge_i \rangle = \frac{1}{1 + \frac{\lambda}{2}\mathbb{E}[f''_{12}]\sqrt{N}\langle x^2, Rx^2 \rangle} \left( \langle x^2, Re_i \rangle - \sqrt{\lambda}\mathbb{E}[f'_{12}]\langle x^2, Rx \rangle\langle x, Ge_i \rangle \right). \tag{69}$$

We notice that $\langle x^2, Rx^2 \rangle = \mathcal{O}(N^{-1})$, $\langle x^2, Re_i \rangle = \mathcal{O}(N^{-2/3})$, and $\langle x^2, Rx \rangle = \mathcal{O}(N^{-2/3})$ from (67). Thus, from the a priori estimate $\langle x, Ge_i \rangle = \mathcal{O}(N^{1/6})$, we obtain that $\langle x^2, Ge_i \rangle = \mathcal{O}(N^{-1/2})$. Plugging it back to (68), we now have that $\langle x, Ge_i \rangle = \mathcal{O}(N^{-1/6})$.

So far, we have seen that

$$\langle x, Ge_i \rangle = \mathcal{O}(N^{-1/6}), \quad \langle x^2, Ge_i \rangle = \mathcal{O}(N^{-1/2}).$$

Plugging these back into (66), we finally obtain that $|R_{ii} - G_{ii}| = \mathcal{O}(N^{-1/3})$. In particular,

$$|\operatorname{Tr} R - \operatorname{Tr} G| = \mathcal{O}(N^{2/3}). \tag{70}$$

To prove the desired theorem, we consider

$$\operatorname{Im} \operatorname{Tr} R = \sum_{i=1}^{N} \frac{\eta}{(\mu_i(V) - \tau)^2 + \eta^2},$$

where $\mu_1(V) \geq \cdots \geq \mu_N(V)$ are the eigenvalues of $V$. We now find an upper bound for $\operatorname{Im}\operatorname{Tr}R$ by partitioning the eigenvalues $\{\mu_i(V)\}$ into the following four groups:

$$E_1 = \{\mu_i(V) : 0 \leq \mu_1(V) - \mu_i(V) \leq N^{-2/3}\}, \quad E_2 = \{\mu_i(V) : N^{-2/3} < \mu_1(V) - \mu_i(V) \leq N^{-5/9}\},$$
$$E_3 = \{\mu_i(V) : N^{-5/9} < \mu_1(V) - \mu_i(V) \leq N^{-2/9}\}, \quad E_4 = \{\mu_i(V) : \mu_1(V) - \mu_i(V) > N^{-2/9}\}.$$

From the rigidity of eigenvalues $\mu_i(V)$, we have that

$$|E_1| = \mathcal{O}(1), \quad |E_2| = \mathcal{O}(N^{1/6}), \quad |E_3| = \mathcal{O}(N^{1/3}), \quad |E_4| = O(N).$$

Recall that we are assuming $\eta = N^{-5/6}$ and $\tau - \mu_1(V) = \xi \geq N^{-3/4} \gg \eta$. Then, for any eigenvalues in $E_1$,

$$(\mu_i(V) - \tau)^2 + \eta^2 \geq N^{-3/2} + \eta^2 \geq N^{-3/2}.$$

For the eigenvalues in $E_2$,

$$(\mu_i(V) - \tau)^2 + \eta^2 \geq (\mu_i(V) - \mu_1(V))^2 \geq N^{-4/3},$$

and similar estimates hold for the eigenvalues in $E_3$ and $E_4$, respectively. Thus, we get

$$
\begin{aligned}
\operatorname{Im}\operatorname{Tr}R &= \left(\sum_{E_1} + \sum_{E_2} + \sum_{E_3} + \sum_{E_4}\right) \frac{\eta}{(\mu_i(V) - \tau)^2 + \eta^2} \\
&= \mathcal{O}(N^{-\frac{5}{6}}/N^{-\frac{3}{2}}) + \mathcal{O}(N^{\frac{1}{6}}N^{-\frac{5}{6}}/N^{-\frac{4}{3}}) + \mathcal{O}(N^{\frac{1}{3}}N^{-\frac{5}{6}}/N^{-\frac{10}{9}}) + O(N^1 N^{-\frac{5}{6}}/N^{-\frac{4}{9}}) \\
&= \mathcal{O}(N^{\frac{2}{3}}).
\end{aligned}
\tag{71}
$$

Now, suppose that $\mu_1(H) \in [\tau, \tau + \eta]$. Then, with $z = \tau + i\eta$,

$$\operatorname{Im}\operatorname{Tr}G = \sum_{i=1}^{N} \frac{\eta}{(\mu_i(H) - \tau)^2 + \eta^2} \geq \frac{\eta}{(\mu_1(H) - \tau)^2 + \eta^2} \geq \frac{1}{2\eta} = \frac{N^{\frac{5}{6}}}{2},$$

which does not hold with overwhelming probability, since $\operatorname{Im}\operatorname{Tr}G = \mathcal{O}(N^{2/3})$ from (70) and (71). Considering $O(N^{1/3+\epsilon})$ such intervals, we can conclude that

$$\mu_1(H) \notin [\mu_1(V) + N^{-3/4}, \mu_1(V) + N^{-1/2+\epsilon}]$$

with overwhelming probability.

Finally, adapting the strategy of the proof of Theorem 2.6 in the supercritical case, we can prove that $|\mu_1(H) - \mu_1(V)| \prec N^{-1/2}$. This completes the proof of the desired theorem. $\square$

## C.2. Proof of the Main Theorem - Subcritical Case

We begin by introducing a result analogous to Proposition 3.3.

**Proposition C.1.** *Let $V(t)$ be the matrix defined in (16) and denote by $\mu_1(V(t))$ the largest eigenvalue of $V(t)$. Fix $\epsilon > 0$. Let $E \in \mathbb{R}$ such that $|E - 2| \leq N^{-2/3+\epsilon}$. Let $E_+ := 2 + 2N^{-2/3+\epsilon}$ and define $\chi_E := \chi_{[E,E_+]}$. Set $\eta_1 := N^{-2/3-\epsilon/2}$ and $\eta_2 := N^{-2/3-3\epsilon}$. Let $K$ and $\theta_\eta$ be defined as in Proposition 3.3. Then, for any $D > 0$,*

$$\mathbb{E}[K(\operatorname{Tr}(\chi_E * \theta_{\eta_2})(H))] > \mathbb{P}(\mu_1(H(t)) \leq E - \eta_1) - N^{-D}$$

*and*

$$\mathbb{E}[K(\operatorname{Tr}(\chi_E * \theta_{\eta_2})(H))] < \mathbb{P}(\mu_1(H(t)) \leq E - \eta_1) + N^{-D}$$

*for any sufficiently large $N$.*

*Proof.* The proof is standard; see, e.g., Proposition 7.1 in (Lee & Schnelli, 2018), with straightforward adjustments by applying the results in (Ajanki et al., 2017). $\square$

Next, we introduce and prove the Green function comparison theorem, analogous to Proposition 3.4.

**Proposition C.2.** *Let $\epsilon > 0$ and set $\eta = N^{-\frac{2}{3}-\epsilon}$. Let $E_1, E_2 \in \mathbb{R}$ satisfy*

$$|E_\ell - 2| \leq N^{-\frac{2}{3}+\epsilon} \quad (\ell = 1, 2).$$

*Let $F : \mathbb{R} \to \mathbb{R}$ be a smooth function satisfying*

$$\max |F^{(m)}(x)| \leq C_m (1 + |x|)^C \qquad (m = 0, 1, 2, 3, 4)$$

*for some constants $C_m > 0$. Then, for any sufficiently small $\epsilon > 0$, there exists $\delta > 0$ such that*

$$\left| \mathbb{E} F \left( \operatorname{Im} \int_{E_1}^{E_2} \operatorname{Tr} R(1, x + \mathrm{i}\eta) \mathrm{d}x \right) - \mathbb{E} F \left( \operatorname{Im} \int_{E_1}^{E_2} \operatorname{Tr} R(0, x + \mathrm{i}\eta) \mathrm{d}x \right) \right| \leq N^{-\delta}. \tag{72}$$

The proof of C.2 is almost a verbatim copy of the proof of 3.4. We nevertheless present here a detailed proof of C.2 for completeness.

*Proof of Proposition C.2.* Fix $x \in [E_1, E_2]$. For simplicity, set

$$R \equiv R(t, x + \mathrm{i}\eta), \quad X \equiv \operatorname{Im} \int_{E_1}^{E_2} \operatorname{Tr} R(t, x + \mathrm{i}\eta) \mathrm{d}x.$$

Then, $X = \mathcal{O}(N^{C\epsilon})$, $F^{(m)}(X) = \mathcal{O}(N^{C\epsilon})$ for some constant $C$. (Throughout the proof, we use $C$ to denote positive constants independent of $N$, whose value may change from line to line.) Differentiating $F(X)$ with respect to $t$, we find the following formula, anaogous to (46):

$$\frac{\mathrm{d}}{\mathrm{d}t} \mathbb{E} F(X) = -\operatorname{Im} \int_{E_1}^{E_2} \mathbb{E} \left[ F'(X) \sum_{i,j,k} \frac{C_1^V \sqrt{N} x_j x_k + C_2^V N x_j^2 x_k^2}{2(1 + C_1^V t \sqrt{N} x_j x_k + C_2^V t N x_j^2 x_k^2)} V(t)_{jk} G_{ij} G_{ki} \right] \mathrm{d}x. \tag{73}$$

Applying Stein's method on the event

$$\Omega_\epsilon = \{ \max_{i,j} |R_{ij} - m_{sc} \delta_{ij}| < N^{-\frac{1}{3}+\epsilon} \text{ for all } x \in [E_1, E_2] \},$$

we expand the right-hand side of (73). In the expansion, the terms involving the third or higher cumulants are negligible in the sense that it is $O(N^{-\frac{1}{3}+C\epsilon})$, which can be proved by performing the naive power counting. (Alternatively, we can rely on the edge universality for Wigner-type matrices to assume that the entries of $V$ are Gaussian for which the third or higher cumulants vanish; see Theorem 1.16 in (Ajanki et al., 2017) and the remark below it.)

From the power counting argument as in the proof of 3.4, on $\Omega_\epsilon$, we find that

$$\begin{aligned}
&\frac{\mathrm{d}}{\mathrm{d}t} \mathbb{E} F(X) \\
&= -\operatorname{Im} \int_{E_1}^{E_2} \sum_{i,j,k} \mathbb{E}[(V(t)_{jk})^2] \frac{C_1^V \sqrt{N} x_j x_k + C_2^V N x_j^2 x_k^2}{2(1 + C_1^V t \sqrt{N} x_i x_j + C_2^V t N x_i^2 x_j^2)} \mathbb{E} \left[ \frac{\partial}{\partial V(t)_{jk}} (F'(X) R_{ij} R_{ki}) \right] \mathrm{d}x \\
&\quad + \mathcal{O}(N^{-\frac{1}{3}+C\epsilon}) \\
&= -\frac{1}{2N} \operatorname{Im} \int_{E_1}^{E_2} \sum_{i,j,k} (C_1^V \sqrt{N} x_j x_k + C_2^V N x_j^2 x_k^2) \mathbb{E} \left[ F''(X) \frac{\partial X}{\partial V(t)_{jk}} R_{ij} R_{ki} + F'(X) \frac{\partial (R_{ij} R_{ki})}{\partial V(t)_{jk}} \right] \mathrm{d}x \\
&\quad + \mathcal{O}(N^{-\frac{1}{3}+C\epsilon}).
\end{aligned}$$

(In the rest of the proof, we always assume that $\Omega_\epsilon$ holds.) We notice that

$$\frac{\partial X}{\partial V(t)_{jk}} = -\operatorname{Im} \int_{E_1}^{E_2} \sum_a (R_{aj} R_{ka} + R_{ak} R_{ja}) \mathrm{d}x = O(N^{-\frac{1}{3}+C\epsilon})$$

from the power counting, and also

$$\frac{1}{N} \int_{E_1}^{E_2} \sum_{i,j,k} (C_1^V \sqrt{N} x_j x_k + C_2^V N x_j^2 x_k^2) \mathbb{E}\left[F''(X) \frac{\partial X}{\partial V(t)_{jk}} R_{ij} R_{ki}\right] \mathrm{d}x = O(N^{-\frac{1}{3}+C\epsilon}).$$

For the main term, which coincides with (47), we apply the same expansion as in the proof of 3.4. Then, we can find that

$$\frac{1}{N} \int_{E_1}^{E_2} \sum_{i,j,k} C_2^V N x_j^2 x_k^2 \mathbb{E}\left[F'(X) R_{ij} R_{kk} R_{ji}\right] \mathrm{d}x = O(N^{-\frac{1}{3}+C\epsilon}).$$

For the remaining term,

$$\frac{1}{N} \int_{E_1}^{E_2} \sum_{i,j,k} C_1^V \sqrt{N} x_j x_k \mathbb{E}\left[F'(X) R_{ij} R_{kk} R_{ji}\right] \mathrm{d}x = O(N^{-\frac{1}{3}+C\epsilon})$$

by splitting $R_{kk}$ into $m_{sc}$ and $(R_{kk} - m_{sc})$ as in the proof of 3.4.

From the estimates above, we find that $\frac{\mathrm{d}}{\mathrm{d}t}\mathbb{E}F(X) = O(N^{-\frac{1}{3}+C\epsilon})$ for any sufficiently small $\epsilon > 0$. Integrating it from $t = 0$ to $t = 1$, we can complete the proof of Proposition C.2. $\square$

Finally, we prove the second part of Theorem 2.5.

*Proof of Theorem 2.5 - Subcritical case.* Proceeding as in the proof of Theorem 2.5 in Appendix B.3, we can prove that the limiting distribution of $\mu_1(\widetilde{M})$ coincides with that of $\mu_1(V)$. Since $V$ is a Wigner matrix, it is well-known that the fluctuation of $\mu_1(\widetilde{M})$ is given by the GOE Tracy–Widom distribution. (See, e.g., Theorem 1.2 in (Lee & Yin, 2014).) This proves the desired theorem. $\square$

## D. Discussion on Different Scaling for the Effective SNR

In this Appendix, we analyze the case $\lambda_e = 0$ and $\lambda = \lambda_0 \sqrt{N}$, considered in Remark 2.8, more in detail. Recall the short-handed notation in (13) and the assumption that $\mathbb{E}[f'_{12}] = 0$, $\mathbb{E}[f''_{12}] \neq 0$. For simplicity, we further assume the i.i.d. prior, where $\sqrt{N}x_i$ are i.i.d. with mean 0 and variance 1, and all moments of $\sqrt{N}x_i$ are finite. While we discuss mainly the supercritical case in this appendix, the subcritical case can also be handled in a similar manner.

### D.1. Approximation by a Spiked Random Matrix

From Taylor's expansion,

$$
\begin{aligned}
\widetilde{M}_{ij} &= \sum_{\ell=0}^{4} \frac{\lambda^{\ell/2} N^{(\ell-1)/2}}{\ell!} f_{ij}^{(\ell)} x_i^\ell x_j^\ell + O(N^{13/4} x_i^5 x_j^5) \\
&= \frac{f_{ij}}{\sqrt{N}} + \sum_{\ell=1}^{4} \frac{\lambda_0^{\ell/2} N^{(3\ell-2)/4}}{\ell!} (f_{ij}^{(\ell)} - \mathbb{E}[f_{ij}^{(\ell)}]) x_i^\ell x_j^\ell + \sum_{\ell=2}^{4} \frac{\lambda_0^{\ell/2} N^{(3\ell-2)/4}}{\ell!} \mathbb{E}[f_{ij}^{(\ell)}] x_i^\ell x_j^\ell + O(N^{13/4} x_i^5 x_j^5) \\
&=: H_{ij} + O(N^{13/4} x_i^5 x_j^5).
\end{aligned}
\tag{74}
$$

Note that $H$ in the right-side of (74) is different from the matrix $H$ defined in 3.1, but we use the notation only in this appendix to make the comparison between the case $\lambda_e = 0$ and the case $\lambda_e \neq 0$. As in Appendix B.3, the Frobenius norm of $(\widetilde{M} - H)$ is bounded by

$$\|\widetilde{M} - H\| = \mathcal{O}\left(N^{13/4} \Big(\sum_{i,j} x_i^{10} x_j^{10}\Big)^{1/2}\right) = \mathcal{O}(N^{-3/4}),$$

which will be negligible when compared to the fluctuation of the largest eigenvalue. This in particular shows that we can use $H$ as an approximation to $\widetilde{M}$.

As a next step, we denote the noise matrix in $H$ by $V$, whose entries are

$$V_{ij} = \frac{f_{ij}}{\sqrt{N}} + \sum_{\ell=1}^{4} \frac{\lambda_0^{\ell/2} N^{(3\ell-2)/4}}{\ell!} (f_{ij}^{(\ell)} - \mathbb{E}[f_{ij}^{(\ell)}]) x_i^\ell x_j^\ell.$$

Again, we use the $V$ as in the equation above only in this appendix to ease the notation when comparing the cases $\lambda_e = 0$ and $\lambda_e \neq 0$. The matrix $V$ is a Wigner-type matrix with the variance

$$\mathbb{E}[(V_{ij})^2] = \frac{1}{N} \left( 1 + C_1^V N^{3/4} x_i x_j + C_2^V N^{3/2} x_i^2 x_j^2 + C_3^V N^{9/4} x_i^3 x_j^3 + C_4^V N^3 x_i^4 x_j^4 \right) + \mathcal{O}(N^{-9/4}).$$

Here, we can notice that the variance of $V_{ij}$ in this case is significantly larger than that in the case $\lambda_e \neq 0$, computed in (15). Indeed, if we compute the sum of $\mathbb{E}[(V_{ij})^2]$, we find

$$\sum_{i,j} \mathbb{E}[(V_{ij})^2] = N + C_2^V \sqrt{N} + O(1).$$

Note that

$$C_2^V = \mathbb{E}[f_{12}^2] + \mathbb{E}[f_{12} f_{12}''] - \mathbb{E}[f_{12}]\mathbb{E}[f_{12}''].\tag{75}$$

Since the sum of the variances of the entries of a Wigner matrix is $N + O(1)$, it means that the limiting empirical distribution of $V$ is not exactly given by the semicircle law but it is stretched by the factor of $(1 + C_2^V N^{-1/2})^{-1/2}$. To compensate this factor, we introduce the interpolation matrix defined by

$$V(t)_{ij} = \left(1 + C_2^V t N^{-1/2}\right)^{-1/2} \sqrt{1 + \sum_{\ell=1}^{4} C_\ell^V t N^{3\ell/4} x_i^\ell x_j^\ell} \frac{V_{ij}}{\sqrt{N\mathbb{E}[(V_{ij})^2]}}.$$

Note that

$$\mathbb{E}[(V(t)_{ij})^2] = \frac{1}{N} + C_1^V t N^{-1/4} x_i x_j + C_2^V t \sqrt{N} \left( x_i^2 x_j^2 - \frac{1}{N^2} \right) + C_3^V t N^{9/4} x_i^3 x_j^3 - C_1^V C_2^V t^2 N^{1/4} x_i x_j + \mathcal{O}(N^{-2}).$$

From the definition of $V(t)$, we have

$$V = \left(1 + C_2^V N^{-1/2}\right)^{1/2} V(1).\tag{76}$$

## D.2. Local Laws

We now continue to prove the local laws for $V(t)$, the estimates corresponding to the statements in Lemma B.3 and Remark B.4. Following the proof of Lemma B.3 in Appendix B.2, with the ansatz $m_i(t,z) = m_{sc}(z) + s_i(t,z)$, we find that

$$(z + m_{sc})s_i + \frac{m_{sc} + s_i}{N} \sum_j s_j + (m_{sc} + s_i) \frac{C_1^V t}{N^{1/4}} \sum_j (m_{sc} + s_j) x_i x_j$$

$$+ (m_{sc} + s_i) C_2^V t \sqrt{N} \sum_j (m_{sc} + s_j)(x_i^2 x_j^2 - \frac{1}{N^2})$$

$$= (z + m_{sc})s_i + \frac{m_{sc} + s_i}{N} \sum_j s_j + (m_{sc} + s_i) \frac{C_1^V t}{N^{1/4}} \sum_j (m_{sc} + s_j) x_i x_j \tag{77}$$

$$+ (m_{sc} + s_i) C_2^V t \sqrt{N} m_{sc}(x_i^2 - \frac{1}{N}) + (m_{sc} + s_i) C_2^V t \sqrt{N} \sum_j s_j(x_i^2 x_j^2 - \frac{1}{N^2}) = \mathcal{O}(N^{-1}).$$

which corresponds to the estimate in (34). Summing the left side of (77) over the index $i$ and dividing it by $N$, we find that

$$(z + 2m_{sc}) \left( \frac{1}{N} \sum_i s_i \right) + \left( \frac{1}{N} \sum_i s_i \right)^2 + \frac{C_1^V t}{N^{5/4}} \left( \sum_i (m_{sc} + s_i) x_i \right)^2$$

$$+ \frac{2m_{sc} C_2^V t}{\sqrt{N}} \left( \sum_i s_i(x_i^2 - \frac{1}{N}) \right) + \frac{C_2^V t}{\sqrt{N}} \sum_{i,j} s_i s_j(x_i^2 x_j^2 - \frac{1}{N^2}) = \mathcal{O}(N^{-1}),\tag{78}$$

which gives $\frac{1}{N} \sum_i s_i = \mathcal{O}(N^{-1/4})$ by naive power counting. Plugging it back into (77), we also find that $s_i = \mathcal{O}(N^{-1/4})$ as well. (Note that these estimates are weaker than the ones we had in Appendix B.2.) By following the bootstrapping argument in Appendix B.2, we can improve the estimates to $\frac{1}{N} \sum_i s_i = \mathcal{O}(N^{-3/4})$ and $s_i = \mathcal{O}(N^{-1/2})$. Finally, by bootstrapping again, we find $\frac{1}{N} \sum_i s_i = \mathcal{O}(N^{-1})$ and $s_i = \mathcal{O}(N^{-3/4})$. Now, combining the estimate $s_i = \mathcal{O}(N^{-3/4})$ with (33), we can conclude that the entrywise local law (32) and the anisotropic local law (37) hold for the current case.

With the entrywise local law and the anisotropic local law for $R$, the resolvent of $V$, we can also prove the local law for $G$, the resolvent of $H$, as in the proof of Proposition 3.2 in Appendix B.2. Here, we have

$$H(t)_{ij} = V(t)_{ij} + \left(1 + C_2^V t N^{-1/2}\right)^{-1/2} \left(\frac{\lambda_0 N}{2} \mathbb{E}[f_{ij}'']x_i^2 x_j^2 + \frac{\lambda_0^{3/2} N^{7/4}}{6} \mathbb{E}[f_{ij}''']x_i^3 x_j^3 + \frac{\lambda_0^2 N^{5/2}}{24} \mathbb{E}[f_{ij}^{(4)}]x_i^4 x_j^4\right),$$

and in particular, $H(0)$ is a rank-3 spiked Wigner matrix and

$$H = \left(1 + C_2^V N^{-1/2}\right)^{1/2} H(1).$$

### D.3. Green Function Comparison

With the local laws for $R$, we can prove a statement analogous to Propositions 3.3 and 3.4. Indeed, the proof of Proposition 3.3 does not change in the current case. To prove Proposition 3.4, we can proceed as in Appendix B.2 and find that the most part of the proof remain unchanged, except that the error bound $O(N^{-\frac{1}{2}+C\epsilon})$ changes to $O(N^{-\frac{1}{4}+C\epsilon})$. The only nontrivial term that should be considered separately is

$$\frac{1}{N} \int_{E_1}^{E_2} \sum_{i,j,k} C_2^V N(x_j^2 x_k^2 - \frac{1}{N^2}) \mathbb{E}\left[F'(X) G_{ij} G_{kk} G_{ji}\right] dx,$$

which is $O(N^{C\epsilon})$ from the naive power counting. To obtain a better estimate for this term, we adapt the idea in (48) to find

$$\frac{1}{N} \int_{E_1}^{E_2} \sum_{i,j,k} C_2^V N(x_j^2 x_k^2 - \frac{1}{N^2}) \mathbb{E}\left[F'(X) G_{ij} G_{kk} G_{ji}\right] dx$$

$$= \frac{1}{N} \int_{E_1}^{E_2} \sum_{i,j,k} C_2^V N(x_j^2 x_k^2 - \frac{1}{N^2}) \mathbb{E}\left[F'(X) G_{ij} m_{sc} G_{ji}\right] dx$$

$$\frac{1}{N} \int_{E_1}^{E_2} \sum_{i,j,k} C_2^V N(x_j^2 x_k^2 - \frac{1}{N^2}) \mathbb{E}\left[F'(X) G_{ij} (G_{kk} - m_{sc}) G_{ji}\right] dx.$$

The second term in the right side of the equation above is $O(N^{-\frac{1}{2}+C\epsilon})$ by the power counting, where we use the local law for $G_{kk} - m_{sc}$. The first term in the right side is further decomposed into

$$\frac{1}{N} \int_{E_1}^{E_2} \sum_{i,j,k} C_2^V N(x_j^2 x_k^2 - \frac{1}{N^2}) \mathbb{E}\left[F'(X) G_{ij} m_{sc} G_{ji}\right] dx$$

$$= \frac{1}{N} \int_{E_1}^{E_2} \sum_{i,j} C_2^V N(x_j^2 - \frac{1}{N}) \mathbb{E}\left[F'(X) G_{ij} m_{sc} G_{ji}\right] dx = \frac{1}{N} \int_{E_1}^{E_2} \sum_{j} C_2^V N(x_j^2 - \frac{1}{N}) \mathbb{E}\left[F'(X) (G^2)_{jj} m_{sc}\right] dx$$

$$= \frac{1}{N} \int_{E_1}^{E_2} \sum_{j} C_2^V N(x_j^2 - \frac{1}{N}) \mathbb{E}\left[F'(X) ((G^2)_{jj} - m_{sc}') m_{sc}\right] dx + \frac{1}{N} \int_{E_1}^{E_2} \sum_{j} C_2^V N(x_j^2 - \frac{1}{N}) \mathbb{E}\left[F'(X) m_{sc}' m_{sc}\right] dx.$$

Note that the second term in the right side of the equation above vanishes due to the summation of $(x_j^2 - N^{-1})$ over the index $j$. To estimate the first term in the right side, we notice that $G^2(z) = G'(z)$. Thus, by applying the Cauchy integral formula for the derivative of $G$ and $m_{sc}$ along the square contour centered at $z$ with the sidelength $\eta$, together with a lattice argument on the contour, we can prove that $(G^2)_{jj} - m_{sc}' = \mathcal{O}(N^{-1/2})$. (The proof of the local law for the derivative, based on the Cauchy integral formula and the lattice argument is standard in random matrix theory; see, e.g., Appendix B of (Chung & Lee, 2022).)

### D.4. Fluctuation of the Largest Eigenvalue

Recall that the largest spike of $H(t)$ is $(1 + o(1))(\lambda_0 N/2)\mathbb{E}[f''_{ij}]\boldsymbol{x}^2(\boldsymbol{x}^2)^T$. If we let $\mathbb{E}[x_i^4] = w_4$, the matrix norm of the largest spike of $H(t)$ converges to $(\lambda_0 w_4/2)\mathbb{E}[f''_{12}]$.

Following the discussion in the previous appendices, we can conclude that the fluctuation of $\mu_1(H(1))$ coincides with that of $\mu_1(H(0))$, and thus it satisfies

$$
N^{1/2}\left(\mu_1(H(1)) - \left(\sqrt{\widetilde{\lambda}_e} + \frac{1}{\sqrt{\widetilde{\lambda}_e}}\right)\right) \to \mathcal{N}(0, \sigma^2)
$$

with $\sigma^2 = 2(\widetilde{\lambda}_e - 1)/\widetilde{\lambda}_e$, where the effective SNR $\widetilde{\lambda}_e$ is given by

$$
\widetilde{\lambda}_e = (\lambda_0^2 w_4^2/4)\mathbb{E}[f''(\sqrt{N}W_{12})]^2].
$$

As the last step, we need to multiply the statement above by the factor $(1 + C_2^V N^{-1/2})^{1/2}$, introduced in the definition of $V(t)$; see (76). We can thus conclude that

$$
N^{1/2}\left(\mu_1(\widetilde{M}) - \left(\sqrt{\widetilde{\lambda}_e} + \frac{1}{\sqrt{\widetilde{\lambda}_e}}\right)\right) \Rightarrow \mathcal{N}(\widetilde{m}, \sigma^2), \tag{79}
$$

where $\sigma^2 = 2(\widetilde{\lambda}_e - 1)/\widetilde{\lambda}_e$ and the mean of the limiting distribution is given by

$$
\frac{C_2^V}{2}\left(\sqrt{\widetilde{\lambda}_e} + \frac{1}{\sqrt{\widetilde{\lambda}_e}}\right).
$$

The argument we used in Appendix D also applies to the subcritical case, which shows that the largest eigenvalue converges to 2 and its fluctuation converges to the GOE Tracy–Widom distribution after it is shifted by $C_2^V N^{-1/2}$. Note that the deterministic shift $C_2^V N^{-1/2}$ is much larger than the size of the Tracy–Widom fluctuation, which is of order $N^{-2/3}$.

Finally, we remark that the argument in Appendix D can be naturally adapted to a more general case. If we let

$$
k_f := \inf\{k \in \mathbb{Z}^+ : \mathbb{E}[f^{(k)}(\sqrt{N}W_{ij})] \neq 0\},
$$

then by following the argument in Appendix D, it can be readily checked that the transition happens in the scaling $\lambda = \lambda_0 N^{\frac{1}{2}(1 - \frac{1}{k_f})}$. (Note that we had assumed $k_f = 2$.) Moreover, we can also prove that the BBP transition in the level of the fluctuation also can be proved in a similar manner, though the proof would require more involved analysis with more levels of bootstraps. We also remark that an appropriate 'correction factor', which was $(1 + C_2^V N^{-1/2})^{1/2}$ for the case $k_f = 2$, should be applied to find the deterministic shift for both the supercritical and the subcritical cases.

