# OpenReview forum: "Fluctuations of the largest eigenvalues of transformed spiked Wigner matrices"
_ICML.cc/2025/Conference — ICML 2025 poster_

### Official Review · Reviewer_RXhe · 2025-03-02

**Overall Recommendation:** 3

**Summary:**

This work investigates the asymptotic properties of the largest eigenvalue of entry-wise, non-linear transformations of spiked Wigner matrices. Its main result is to prove a BBP-like result: below a certain critical SNR (which is explicitly given in terms of the original SNR), the top eigenvalue has asymptotic Gaussian fluctuations, while below it it follows a Tracy-Widom law. The result is proven using an interpolation + resolvent comparison argument that requires proving a local law for this class of matrices, which is of independent interest. Numerical simulations are also given to illustrate the result.

**Claims And Evidence:**

Yes, all claims in this work are rigorous mathematical statements, and the proofs are provided.

**Essential References Not Discussed:**

To my best knowledge, the first rigorous universality results in this direction appeared in (Krzakala et al., 2016; Dia et al., 2016; Lelarge & Miolane 2017), who proved the universality of the asymptotic mutual information with respect to the noise channel conjectured by Lesieur et al., (2015). These are key related reference from which (Perry et al., 2018) build upon.

More recently, (Mergny et al. 2024) extended these universality results to cases in which the Fisher information is zero, requiring an analysis of different SNR regimes.

On the related literature side, a recent result by Dandi et al., (2024) proved rigorous deterministic equivalents for non-linear spiked models coming from single gradient steps.

- [Krzakala et al., 2016] Krzakala, Florent, Jiaming Xu, and Lenka Zdeborová. "Mutual information in rank-one matrix estimation." In 2016 IEEE Information Theory Workshop (ITW), pp. 71-75. IEEE, 2016.

- [Dia et al., 2016] Dia, Mohamad, Nicolas Macris, Florent Krzakala, Thibault Lesieur, and Lenka Zdeborová. "Mutual information for symmetric rank-one matrix estimation: A proof of the replica formula." Advances in Neural Information Processing Systems 29 (2016).

- [Lelarge & Miolane 2017] Lelarge, Marc, and Léo Miolane. "Fundamental limits of symmetric low-rank matrix estimation." In Conference on Learning Theory, pp. 1297-1301. PMLR, 2017.

- [Mergny et al. 2024] Mergny, Pierre, Justin Ko, Florent Krzakala, and Lenka Zdeborová. "Fundamental limits of non-linear low-rank matrix estimation." Proceedings of Thirty Seventh Conference on Learning Theory, PMLR 247:3873-3873, 2024.

- [Dandi et al., 2024] Dandi, Yatin, Luca Pesce, Hugo Cui, Florent Krzakala, Yue M. Lu, and Bruno Loureiro. "A random matrix theory perspective on the spectrum of learned features and asymptotic generalization capabilities." arXiv preprint arXiv:2410.18938 (2024).

**Experimental Designs Or Analyses:**

N/A.

**Methods And Evaluation Criteria:**

N/A.

**Other Comments Or Suggestions:**

- Page 1, below eq. (1), $H$ should read $W$.

**Other Strengths And Weaknesses:**

Overall, the paper is well-written and easy to read, and the result is solid. However, I wonder about the choice of venue, since as a "pure" RMT paper it will only interest a small part of the ICML community. Maybe the authors could have put more effort into motivating how the spiked estimation problem pops up in ML, giving some more concrete examples and context. Otherwise, I think it is a good RMT paper.

**Questions For Authors:**

- In the first paragraph of "1.1. Contributions" and in eq. (5), should the effective SNR be $f'$ instead? Maybe I am missing something, but if this is correct, how do you reconcile with the result of (Lesieur et al., 2015) discussed in eq. (3)?

- On the ML side, there has been a wave of recent interest in non-linear transformations of spiked Wigner/Wishart matrices, mostly motivated by the fact that, asymptotically, a single step of GD for a two-layer neural network trained with the square loss looks like a matrix in this ensemble. This characterization was used to prove the benefit of feature learning in several works (Ba et al. 2022; Moniri et al., 2023; Cui et al., 2024; Dandi et al., 2024).

  However, an important point in these works is that feature learning is only beneficial when the scaling of the spike (SNR here) is very large - i.e. scales with $N$. This regime is way above the BBP transition. Would the theorem here apply? If not, could the argument be generalized to cover this case?

- What are the main technical challenges to extend the proof to the non-symmetric, spiked Wishart case?

- How important is the assumption that the spike is dense $||x||_{2}^{2}=1$? The universality proofs for the mutual information of [Krzakala et al. 2016; Mergny et al. 2024] only require the support of the spike distribution to be finite.

**Relation To Broader Scientific Literature:**

The universality of entry-wise non-linear functions of spiked Wigner matrices is the topic of several previous works, since it has important consequences for statistical inference. Different flavours of this result exist in the literature, the one most relevant to this work being Perry et al., (2018), which this work build upon and is thoroughly acknowledged.

However, some relevant important literature is missing - see below.

**Theoretical Claims:**

I skimmed through the proof, and did not spot any issue. The arguments are standard, but require controlling a rank-2 spiked Wigner matrix.

---

> ### Author Rebuttal · Authors · 2025-04-01
>
> We thank the reviewer for the constructive review and helpful feedback with which we can greatly improve our paper. Below, we address the concerns and questions, and also outline some important revisions we will make.
>
> **References**: Thank you for the references that we missed. We will include them in the revision.
>
> **Relevance to the ICML audience**: Thank you for the comment. In the revision, we will add and change a few paragraphs in the introduction of the current manuscript to demonstrate the relevance of our model to the ICML audience. The details can be found in our answer to Reviewer 1 (dwXH); I am sorry that we cannot write all the details in this box due to the length limit,
>
> **Other comments and questions**:
>
> (1) Thank you for pointing out the typos. We will correct them and other typos in the revision. Also, $f$ in the definition in $\lambda_e$ is a typo, which should be $f'$.
>
> (2) Thank you for your comment on the large SNR regime. If $\lambda_e = \omega(1)$, we believe that our result on the supercritical case still holds, i.e., $N^{1/2}(\mu_1 - \sqrt{\lambda_e} - 1/\sqrt{\lambda_e})$ converges to a centered Gaussian distribution with variance $2$. However, to prove it rigorously by adapting the strategy in the current manuscript, we need to find a proper target matrix for which the limiting distribution of the largest eigenvalue is proven, but unfortunately, to our best knowledge, it is not proven. Nevertheless, we would remark that the large ($\omega(1)$) SNR case should be relatively easier to analyze than the $O(1)$ SNR case and the argument in the current manuscript should be applicable to this case without many changes.
>
> We also would remark that one can consider the case where the SNR is large but $\lambda_e = 0$ since $\mathbb{E}[f_{12}'] = 0$. See our answer to Reviewer 1 (dwXH), item **Question on generalization**, for more detailed discussion on this case.
>
> (3) The main technical challenge for the non-symmetric or rectangular case is that the local law is much harder to prove. More specifically, we need to use a self-consistent equation for the resolvent entries analogous to the quadratic vector equation (21), but its analysis is significantly more complicated due to some technical issues. Furthermore, if we consider a multiplicative model considered in (Perry et al., 2018; Jung et al., 2021), then the transformed matrix cannot be well approximated by a spiked random matrix. This means that the strategy in the current manuscript should be changed accordingly.
>
> (4) If we let $y_i$ be i.i.d. with an $N$-independent distribution whose mean is $0$, variance is $1$, and all moments are finite, and set $x_i = y_i/\sqrt{N}$, then Assumption 2.3 is satisfied with overwhelming probability. The prior distribution (of $y_i$) does not need to be compactly supported, so our assumption on the spike is weaker than that in (Mergny et al. 2024) in this sense. Further, the prior distribution may have an atom at $0$, so the spike does not need to be 'dense' or 'flat'.
>
> One possible generalization of our result is to allow a `sparse' spike in the sense that $\max_i |x_i| > N^{-1/2 +\delta}$. With such a sparse spike, the fluctuation of the largest eigenvalue might exhibit another phase transition depending on $\delta > 0$. This is an interesting topic and we would like to consider it in the future.

---

> > ### Comment · Reviewer_RXhe · 2025-04-01
> >
> > I thank the authors for their rebutal. My questions were clarified, and I will keep my score.

---

### Official Review · Reviewer_kzF3 · 2025-03-14

**Overall Recommendation:** 3

**Summary:**

The authors study the largest eigenvalue of a spiked Wigner matrix model under elementwise function transformation. They show that the largest eigenvalue of the transformed matrix undergoes a phase transition when an effective SNR variable is tuned. At high SNR, the find that the largest eigenvalue is distributed as a gaussian around a value computable from the effective SNR. At low SNR, it is a Tracy-Widom distribution around 2. The authors establish this via proof and support their claims via numerics.

**Claims And Evidence:**

Yes, the claims are well supported

**Essential References Not Discussed:**

N/A

**Experimental Designs Or Analyses:**

Yes, I checked the validity of both the numerical experiments.
The first numerical experiment is not very clear to me. The f used here is defined in terms of the p and it's derivative. The authors should be more concrete about what this function explicitly looks like.

**Methods And Evaluation Criteria:**

Yes, the proposed methods make sense.

**Other Comments Or Suggestions:**

1.) After eq(1) there is a typo. H should be W

2.) In Eq(4) the bracket in f' is not closed

3.) In several points in the paper \lambda_e is defined with f instead of f'. This looks like a typo. If defined with f, the effective SNR will vanish by assumption.

4.) For the sake of completeness, could the authors briefly discuss what happens when \lambda_e = 1

 5.) How are the assumptions 2.3 on x justified? It seems to say that the x vector must be fairly flat. For the community detection case where x_i are proportional to {-1,1} what do these assumptions mean? Does it translate to a statement about the size of the communities?

6.) Do the techniques used here tell us something about the eigenvector corresponding to the largest eigenvalue? For PCA type of applications this is important to know. Will this eigenvector have a large overlap with the spike?

**Other Strengths And Weaknesses:**

See questions

**Questions For Authors:**

N/A (See previous)

**Relation To Broader Scientific Literature:**

The paper basically shows what the statistical fluctuations of the largest eigenvalue looks like in the spiked Wigner model under element wise transformations. In the un-transformed case,  similar results are known and have applications in community detection. This is a useful generalization of this as it allows us to rigorously boost the SNR in these problems.

**Theoretical Claims:**

I checked the proof of the main theorems. Propositions were not checked.

---

> ### Author Rebuttal · Authors · 2025-04-01
>
> We thank the reviewer for the constructive review and helpful feedback with which we can greatly improve our paper. Below, we address the concerns and questions, and also outline some important revisions we will make.
>
> **Experimental design**: Thank you for your comment. We will include the graphs of the density $p$ and the transformation $f$ in the revision. (The precise formulas for $p$ and $f$ can be found in Appendix A.1.) See also the following links for the graphs of the [density](https://www.wolframalpha.com/input?i=Draw+%282.5%2F%5Csqrt%7B2%5Cpi%7D%29+*%5Be%5E%7B-%285%2F2%29%28x-2%2Fsqrt%7B5%7D%29%5E2%7D+%2B+e%5E%7B-%285%2F2%29%28x%2B2%2Fsqrt%7B5%7D%29%5E2%7D%5D) and the [transformation](https://www.wolframalpha.com/input?i=Draw+2.62503*%5B%28x-2%2F%5Csqrt%7B5%7D%29+e%5E%7B-%285%2F2%29%28x-2%2Fsqrt%7B5%7D%29%5E2%7D+%2B+%28x%2B2%2F%5Csqrt%7B5%7D%29+e%5E%7B-%285%2F2%29%28x%2B2%2Fsqrt%7B5%7D%29%5E2%7D%5D%2F%5B+e%5E%7B-%285%2F2%29%28x-2%2Fsqrt%7B5%7D%29%5E2%7D+%2B++e%5E%7B-%285%2F2%29%28x%2B2%2Fsqrt%7B5%7D%29%5E2%7D%5D).
>
> **Other comments and questions**:
>
> (1)-(3) Thank you for pointing out the typos. We will correct them and other typos in the revision. Also, $f$ in the definition in $\lambda_e$ is a typo, which should be $f'$.
>
> (4) If $\lambda_e = 1$, then the largest eigenvalue $\mu_1 (\widetilde{M})$ converges to $2$, but the fluctuation $N^{2/3}( \mu_1 (\widetilde{M}) - 2)$ converges to a distribution different from the GOE Tracy-Widom distribution. To our best knowledge, the limiting distribution involves the stochastic Airy operator and does not have simple formula. We refer to Theorems 1.5 and 1.7 in (Bloemendal and Virag, 2013) for more detail.
>
> (5) If we let $y_i$ be i.i.d. with an $N$-independent distribution whose mean is $0$, variance is $1$, and all moments are finite, and set $x_i = y_i/\sqrt{N}$, then Assumption 2.3 is satisfied with overwhelming probability. (This can be proved by concentration inequalities such as a high-order Markov inequality.) Our assumption on the spike is not much different from the assumptions in other related papers, e.g., (Guionnet et al., 2023; Feldman, 2023). Further, the prior distribution may have an atom at $0$, so the spike does not need to be 'dense' or 'flat'. If we consider the community detection case, Assumption 2.3 is satisfied if $|\{ x_i = 1/\sqrt{N} \}| - |\{ x_i = -1/\sqrt{N} \}| = O(N^{1/2 + \epsilon})$, which implies that the model is (almost) balanced.
>
> (6) If we let $\boldsymbol v$ be the eigenvector corresponding to the largest eigenvalue, it is known that $\langle \boldsymbol v,  \boldsymbol x^2 \rangle \to (1 -1/\lambda_e)$ if $\lambda_e > 1$, and $\langle \boldsymbol v, \boldsymbol x^2 \rangle \to 0$ otherwise. (See Corollary 2.2 in (Guionnet et al., 2023).)

---

### Official Review · Reviewer_dwXH · 2025-03-15

**Overall Recommendation:** 3

**Summary:**

This paper studies the fluctuations of the largest eigenvalues in transformed spiked Wigner matrices and discuss the Baik–Ben Arous–Péché (BBP)-type phase transition arisiring in these problem. While a great deal is known about these matrix, the contribution is the analysis of the asymptotic fluctuation  of the leading eigenvalue, proving that it follows a Gaussian distribution in the supercritical regime (when the effective signal-to-noise ratio is above a critical threshold) and the Tracy–Widom GOE distribution in the subcritical regime.Their approach relies on precise concentration estimates, interpolation arguments, and Green function techniques to track the eigenvalue fluctuations.

**Claims And Evidence:**

The claims made in the paper are well-supported by rigorous mathematical proofs. I have checked --not in detail-- the theoretical derivations and found them correct.

**Essential References Not Discussed:**

The literature review in this paper is well done, and I do not have much to add. The only minor but possible suggestion is related to a question bellow is related to  arXiv:2403.04234, "Fundamental Limits of Non-Linear Low-Rank Matrix Estimation."

**Experimental Designs Or Analyses:**

N/A

**Methods And Evaluation Criteria:**

Since this is a theoretical work, there are no experimental design concerns. The supplementary material, which contains additional proofs, was reviewed.

**Other Comments Or Suggestions:**

See bellow

**Other Strengths And Weaknesses:**

The paper is solid and presents a rigorous mathematical analysis of eigenvalue fluctuations in transformed spiked Wigner matrices. However, it leans heavily towards mathematical random matrix theory (RMT) rather than machine learning, which raises the question of its relevance to the ICML audience. While the results are interesting and technically sound, their direct impact on practical ML applications remains somewhat unclear. The theoretical insights are valuable for understanding spectral methods, but their significance for ICML could be better motivated.

**Questions For Authors:**

My main question concerns a possible generalization. In works such as Guionnet et al. (2023) and Feldman (2023), a slightly different scaling is considered, which is further explored in detail in arXiv:2403.04234.  In this generalization, the transformation starts with  a signal-to-noise ratio scaling as N^{1-1/k_f) , but still ends up with a BBP-type transition for the trasnformed spectral method.  I am curious whether the authors' approach could also yield Tracy–Widom-type fluctuations in this setting. If so, this would significantly enhance the impact and applicability of the results.

**Relation To Broader Scientific Literature:**

The paper builds on the well-established literature on spiked random matrix models and eigenvalue phase transitions, particularly the Baik–Ben Arous–Péché (BBP) transition. It extends results from classical spiked Wigner matrix theory by considering entrywise transformations and their impact on spectral properties, a direction that has been explored in works such as Lesieur et al. (2015), Perry et al. (2018).

The main novelty here is the rigorous characterization of eigenvalue fluctuations under such transformations, demonstrating that the largest eigenvalue follows a Gaussian or Tracy–Widom GOE distribution depending on the effective signal strength. This aligns with recent advances in random matrix theory that analyze the stability and universality of spectral methods under transformations, as seen in both theoretical works and applications in statistical inference and machine learning.

While the paper is mathematically strong, its direct connection to broader ML applications is somewhat limited. Similar ideas have been used in theoretical analyses of feature learning in neural networks and optimization landscapes, suggesting a potential impact beyond pure RMT. The author discusses some of these aides, but  making this connection more explicit could strengthen its relevance to the ICML audience?

**Theoretical Claims:**

The methodology follows standard theoretical techniques in random matrix theory, and the supplementary material provides complete proofs of the results.

---

> ### Author Rebuttal · Authors · 2025-04-01
>
> We thank the reviewer for the constructive review and helpful feedback with which we can greatly improve our paper. Below, we address the concerns and questions, and also outline some important revisions we will make.
>
> **Relevance to the ICML audience**: Thank you for the comment. In the revision, we will incorporate the paragraphs below into the introduction of the current manuscript to demonstrate the relevance of our model to the ICML audience.
>
> Spiked random matrix models play a crucial role in analyzing for many machine learning problems. For instance, in the study of the feature learning in a two-layer neural network, a rank-$1$ perturbation of the initial weight matrix can approximate the updated weight after one gradient step (Ba et al. 2022). This idea is further developed by (Cui et al. 2024; Dandi et. al., 2024), where the spiked Random Features model was considered with first layer weights of the form $X + r \boldsymbol{u}\boldsymbol{v}^T$. Additionally, these models have practical applications in machine learning algorithms, such as an ensemble method for Q-learning in reinforcement learning (Lee et al, 2023).
>
> The spectral properties of spiked random matrix models are important in analyzing various problems in machine learning and statistics, as spectral methods can improve convergence analysis and provides theoretical guarantees. Notable examples include the community detection (Abbe, 2017) and submatrix localization (Butucea et al., 2013). In the context of machine, we refer to (Mondelli and Venkataramanan, 2021) for an application to the approximate message passing algorithm and (Chi et al., 2019) to non-convex optimization.
>
> Entrywise transformed random matrix models provide valuable insights into the study of fundamental limits of detection for the signal from a spiked random matrix, which has been extensively studied by analyzing the eigenvalues (Montanari et al., 2015; Johnstone and Onatski, 2015) or the mutual information and minimum mean squared error (Lesieur et al., 2015; Krzakala et al., 2016; Dia et al., 2016; Lelarge and Miolane, 2017). Further, these models contribute to the theoretical analysis of deep neural network, where entrywise transformations correspond to pointwise nonlinear activation functions (Pennington and Worah, 2017).
>
> **References**: Thank you for the reference of (Mergny et al., 2024) that we missed. We will include it in the revision.
>
> **Question on generalization**: Thank you for the good question. It is also possible to prove similar results with different SNR scaling. In our setting, the index $k_f$ can be defined as $k_f := \inf \\{ k \in \mathbb{Z}^+ : \mathbb{E}[f_{12}^{(k)}] \neq 0 \\}$. Under a mild assumption on $\boldsymbol x$ (e.g., $\sqrt{N} x_i$'s are i.i.d. with an $N$-independent distribution whose mean is $0$, variance is $1$, and all moments are finite), the critical scaling of the SNR is $\lambda \equiv \lambda(N) = \lambda_0 N^{1-1/k_f}$ and the entries of the spiked Wigner matrix are given by $M_{ij} = W_{ij} + \sqrt{\lambda_0} N^{\frac{1}{2}(1-\frac{1}{k_f})} x_i x_j$. Let us consider the case $k_f = 2$ for simplicity. The Taylor expansion in (3) is then $$N^{-\frac{1}{2}} f(\sqrt{N}M_{ij}) \approx \frac{f_{ij}}{\sqrt N} + \sqrt{\lambda_0} N^{1/4} f_{ij}' x_i x_j + \sqrt{\lambda_0} N (f_{ij}''- \mathbb{E}[f_{ij}'']) x_i^2 x_j^2 + \sqrt{\lambda_0} N \mathbb{E}[f_{ij}''] x_i^2 x_j^2.$$ This suggests that the effective SNR $\lambda_e = \lambda_0 \mathbb{E}[f_{12}]^2 \| {\boldsymbol x}^2 \|^4$. To prove the BBP transition, we can proceed as in the current paper with several changes. The most significant change is that the noise matrix $V$ in (14) is now given by $$V_{ij} = \frac{f_{ij}}{\sqrt N} + \sqrt{\lambda_0} N^{1/4} f_{ij}' x_i x_j + \sqrt{\lambda_0} N (f_{ij}''- \mathbb{E}[f_{ij}'']) x_i^2 x_j^2.$$ Due to the change of the noise matrix $V$, we need to adjust several parts of the current paper. The most notable ones are as follows:
>
> (1) The (row) sum of the variances, $\sum_j \mathbb{E}[(V_{ij})^2] = 1+ \Theta(N^{-1/2})$ (which was $1+ \Theta(N^{-1})$). This means that all eigenvalues are rescaled by a (deterministic) factor $1+ \Theta(N^{-1/2})$, or equivalently, all eigenvalues are shifted by $\Theta(N^{-1/2})$. Since the shift is not negligible when compared to the typical size of the fluctuations ($\Theta(N^{-1/2})$ or $\Theta(N^{-2/3})$), our main results should be changed accordingly.
>
> (2) The proof of the local law for $V(t)$, Lemma B.3, and other proofs based on the local law and the resolvent identity, will become more involved. We need to use the bootstrap argument more times, where we first prove a (weak) estimate first and improve it by using the weak estimate itself.
>
> In the revision, we will incorporate the discussion above into the manuscript and add detailed proofs to the appendix.

---

### Decision · Program_Chairs · 2025-05-01

**Decision:**

Accept (poster)

**Comment:**

This paper provides a detailed fluctuation analysis of the largest eigenvalues in spiked Wigner matrices under entrywise nonlinear transformations. The main technical contribution lies in identifying the non-universality of the leading eigenvalue fluctuations in this setting—showing that unlike classical spiked models, the fluctuations depend on the specific form of the transformation function. The reviewers are in broad agreement that this is a valuable contribution to the theory of random matrices with relevance for understanding the behavior of spectral methods in high-dimensional settings. The paper is technically strong and carefully written.

At the same time, reviewers noted that the presentation is somewhat dense and could be improved to make the significance of the results more accessible. In particular, it would help to provide stronger motivation from machine learning contexts where such transformations naturally arise—e.g., in kernel PCA, graph embeddings, or score-based generative models. Similarly, adding a clearer explanation of how the fluctuation results relate to algorithmic questions (e.g., detectability thresholds, inference error) would broaden the paper’s impact. While the results are mathematical, the framing could better bridge the gap to applications that ICML readers care about. The authors are encouraged to clarify these points in the final version. Given the novelty and technical depth, the paper deserves to be accepted.